# ADAPTIVE CONCEPT DISCOVERY FOR INTERPRETABLE FEW-SHOT TEXT CLASSIFICATION

**Lifang ZHENG, Hanmo LIU, Kani CHEN**[*]
The Hong Kong University of Science and Technology
{lzhengah, hliubm}@connect.ust.hk, makchen@ust.hk

## ABSTRACT

Few-shot text classification is a critical real-world task for which Large Language Models (LLMs) have shown great promise. However, their high inference costs and lack of interpretability limit their practical use. While Concept Bottleneck Models (CBMs) offer an efficient and interpretable alternative, their reliance on training surrogate models makes them incompatible with few-shot scenarios. To bridge this gap, we introduce a novel CBM paradigm that relies solely on sample-concept similarity to make predictions. We ensure the effectiveness of our concepts through a prototypical-discriminative dual-level architecture and a dynamic concept refinement mechanism. Extensive experiments show that with as few as 10 training samples, our method surpasses prior CBMs and even achieves performance comparable to LLMs. The code is available at https://github.com/alexiszlf/StructCBM.

## 1 INTRODUCTION

While Large Language Models (LLMs) have set new performance standards for few-shot and zero-shot text classification (Brown et al., 2020; Wei et al., 2022; OpenAI, 2023; Gao et al., 2021; Min et al., 2022), their practical deployment is constrained by significant trade-offs. On one hand, their substantial computational requirements present a major barrier to scalable, cost-effective inference on large datasets (Zhou et al., 2024). On the other hand, the "black-box" nature of their reasoning process raises critical concerns about trustworthiness, particularly in regulated fields such as finance, healthcare, and legal services, where decision transparency is non-negotiable (Bender et al., 2021; Doshi-Velez & Kim, 2017).

To address the challenges of high inference cost and lack of interpretability in LLMs, Concept Bottleneck Models (CBMs) (Koh et al., 2020) offer a promising alternative. As lightweight and inherently interpretable "white-box" models, CBMs ground their predictions in a set of human-understandable concepts, thereby providing both efficiency and transparency. Recognizing this potential, some pioneering studies have explored using the extensive world knowledge of LLMs to help CBMs automatically extract these concepts (Luyten & van der Schaar, 2024; Ludan et al., 2024; Feng et al., 2024; Sun et al., 2025).

However, the application of CBMs, particularly in few-shot scenarios, introduces its own set of challenges. While theoretical work suggests that CBMs are well-suited for few-shot learning by improving sample efficiency (Luyten & van der Schaar, 2024), practical implementations face significant hurdles. Specifically, the limited few-shot data are insufficient to train a reliable prediction layer for mapping concepts to labels. Furthermore, existing LLM-augmented CBMs face a critical trade-off: iterative approaches like TBM and BC-LLM (Ludan et al., 2024; Feng et al., 2024) remain dependent on costly LLMs during inference, while models designed for efficient inference, such as CB-LLMs (Sun et al., 2025), require large datasets for training and are thus ill-suited for few-shot learning. A final limitation is that these models often frame the text-concept similarity as a supervised learning objective, precluding the possibility of iteratively refining the concepts themselves.

To address this challenge, we introduce StructCBM, a novel framework centered around a prototypical-discriminative concept architecture. We first employ a large language model to gen-

---

[*]Corresponding author.

Table 1: Summary on current text concept bottleneck models.

| Method | Few-shot | Inference w/o LLM | Classifier Free |
|---|---|---|---|
| TBM (Ludan et al., 2024) | No | No | No |
| C$^3$M (Tan et al., 2024b) | No | Yes | No |
| SparseCBM (Tan et al., 2024a) | No | Yes | No |
| CB-LLM(Sun et al., 2025) | No | Yes | No |
| CT-CBM(Bhan et al., 2025) | No | Yes | No |
| StructCBM (Ours) | Yes | Yes | Yes |

erate two distinct sets of concepts from a small number of samples: general prototypical concepts to identify a candidate set of labels, and fine-grained discriminative concepts to distinguish between them. The prediction is then made via a lightweight, two-stage process that relies on efficient text embedding similarity, removing the need for the LLM during inference. To ensure the quality of our concepts, the framework is designed as a closed-loop system where misclassifications are fed back to the LLM, which then performs semantic and logical refinements to iteratively tune the concept library. This generate-predict-refine workflow allows our model to remain highly efficient and interpretable while effectively handling the constraints of a few-shot learning environment. Our main contributions are summarized as follows:

- We are the first to apply LLM-augmented concept bottleneck models to few-shot learning, establishing a novel direction that leverages the synergy between the interpretability of CBM and the few-shot reasoning ability of LLM.

- We propose a dual-level, prototypical-discriminative concept architecture that handles few-shot learning directly through sample-concept matching. We further design a generate-predict-refine loop to guarantee the concept effectiveness.

- We demonstrate through extensive experiments that our method achieves performance comparable to black-box baselines and surpasses prior LLM-based CBMs on multiple benchmarks.

## 2 RELATED WORK

**Few-shot Text Classification.** Few-shot text classification aims to build high-performance classifiers using only a small number of labeled samples. Mainstream approaches can be broadly divided into three categories. The first involves fine-tuning pretrained language models (PLMs)(Wei et al., 2022). The second category consists of prompt-based learning methods, which leverage task-specific templates to elicit the prior knowledge of language models. The third is the in-context learning paradigm of large language models (LLMs)(Brown et al., 2020; Min et al., 2022), such as directly using GPT-4 for few-shot prediction. Although these methods have achieved remarkable performance, their decision-making processes are opaque, and the high inference costs of large models limit their application in domains requiring decision auditing or handling large-scale data.

**Concept Bottleneck Models.** To address the black box nature of modern NLP models, research has pursued two primary paths: post hoc and inherent interpretability. Post hoc methods, like feature attribution (e.g., LIME, SHAP)(Ribeiro et al., 2016; Lundberg & Lee, 2017), aim to explain pretrained models, but studies have shown their explanations may not be faithful to the model's actual reasoning. In contrast, inherently interpretable models like the Concept Bottleneck Model (CBM) are transparent by design, forcing predictions through a layer of human-understandable concepts and thus providing a faithful explanation for each decision.

Building on this framework, a recent line of work utilizes LLMs to define concepts for a CBM, while a smaller model performs the final prediction. These methods differ in their concept generation and inference paradigms. Some approaches generate a static concept library for a given task, such as CB-LLMs (Sun et al., 2025) and SparseCBM (Tan et al., 2024a), or expand upon human-provided seed concepts, as in C³M (Tan et al., 2024b). Others introduce dynamic concept discovery. For example, Ludan et al. (2024) introduced TBM, which uses classification errors to iteratively prompt an LLM for new concepts, while Feng et al. (2024) framed this process in a Bayesian framework with BC-LLM, exploring the concept space via MCMC sampling. A critical drawback of these models is their reliance on the LLM during inference. In contrast, methods like CB-LLMs and Sparse-CBM achieve efficient prediction by training an embedding model to match samples to concepts (Sun et al., 2025;

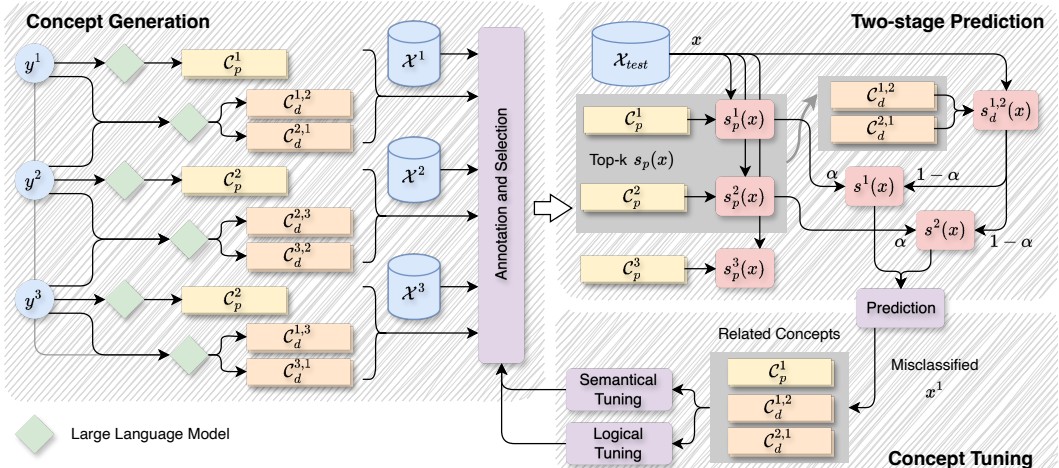

Figure 1: The framework of StructCBM. We first prompt an LLM to generate an initial library of prototypical concepts $\mathcal{C}_p$ and discriminative concepts $\mathcal{C}_d$. Next, we predict with the concept library in a two-stage process: we first select candidate labels by ranking sample similarities with $\mathcal{C}_p$, and then finalize the label through a contrastive comparison among these candidates using $\mathcal{C}_d$. For misclassified samples, we semantically or logically refine the concept library based on their feedback in the concept tuning stage.

Semenov et al., 2024). Other works introduce parallel residual paths to improve performance, as seen in TBM (Ludan et al., 2024) and CT-CBM (Bhan et al., 2025), but this hybrid architecture risks information leakage, potentially compromising the faithfulness of the explanations. A summary of related CBMs for text classification is shown in Tab. 1

## 3 METHODS

In this section, we first introduce the motivation of our prototypical-discriminative concept structure in Sec. 3.1. Then, to exploit the advantages of our concept structure, our workflow is designed in a generate-predict-refine pattern, where are introduced in Sec. 3.2 to Sec. 3.4 and illustrated in Fig. 3.

### 3.1 PRELIMINARY

To establish an interpretable and efficient model, recent CBMs (Sun et al., 2025; Ludan et al., 2024) often decomposes the prediction procedure into two sequential functions $g \circ f(\cdot)$: a concept matching function $f(\cdot) : \mathcal{X} \to \mathbb{R}^{|\mathcal{C}|}$ that maps the input data $\mathcal{X}$ to a set of prepared concepts $\mathcal{C}$ and a white-box prediction model $g(\cdot) : \mathbb{R}^{|\mathcal{C}|} \to \mathbb{R}^{|\mathcal{Y}|}$ that predicts on the label set $\mathcal{Y}$ based on the concept relatedness. This framework is a good substitute for black-box models when there is abundant labeled data, but it would fail to handle few-shot scenarios due to the inability to optimize $g(\cdot)$.

Motivated by this dilemma, we leverage extensive prior knowledge of LLMs to establish an explicit one-to-one correspondence between concepts and labels. This allows us to simplify the prediction function $g(\cdot)$ into a static mapping, thus avoiding classifier training and shifting the entire burden of prediction optimization to the concept representation function $f(\cdot)$.

From a probabilistic perspective, predicting on sample $x$ with a concept library $\mathcal{C}$ can be expressed as $\hat{y} = \arg\max_{y \in \mathcal{Y}} P(y|x, \mathcal{C})$. With few-shot samples and possibly numerous concepts, using parametric methods to directly optimize this distribution is still impossible. Inspired by coarse-to-fine strategies, we re-frame this problem as follows:

$$P(y|x, \mathcal{C}) = \sum_{\mathcal{Y}_k \subset \mathcal{Y}} P(y|\mathcal{Y}_k, x, \mathcal{C}) P(\mathcal{Y}_k|x, \mathcal{C}), \tag{1}$$

where $\mathcal{Y}_k \subset \mathcal{Y}$ contains $k$ candidate labels and $P(y|\mathcal{Y}_k, x, \mathcal{C}) = 0$ when $y \notin \mathcal{Y}_k$. Following this extension, the prediction is transformed into 1) finding top-$k$ candidate labels from $\mathcal{Y}$ that contain the ground-truth and 2) finalizing the top-1 label from $k$ candidates. This decomposition reveals two

distinct sub-problems: a recall-oriented task of generating a high-quality candidate set ($P(\mathcal{Y}_k|x,\mathcal{C})$), and a precision-oriented task of ranking labels within that set ($P(y|\mathcal{Y}_k, x, \mathcal{C})$). Our key insight is that a single, flat concept library is ill-suited to optimally solve both. We therefore propose a two-stage architecture that employs specialized concept sets for each sub-problem: general *Prototypical Concepts* $\mathcal{C}_p$ to ensure high recall in the first stage, and specific *Discriminative Concepts* $\mathcal{C}_d$ to achieve high precision in the second:

- **Prototypical Concept Set** $\mathcal{C}_p$**:** Structured as a union of per-class subsets $\mathcal{C}_p = \bigcup_{y^i \in \mathcal{Y}} \mathcal{C}_p^i$, where each subset $\mathcal{C}_p^i$ is one-to-one correspondent to a label $y^i$ from the label set $\mathcal{Y}$. Each concept $c_p^i \in \mathcal{C}_p^i$ captures a common and representative concept that defines the core identity of class $y^i$.

- **Discriminative Concept Set** $\mathcal{C}_d$**:** Structured as a union of pairwise subsets $\mathcal{C}_d = \bigcup_{y^i \neq y^j} \mathcal{C}_d^{i,j}$. Each concept $c_d^{i,j} \in \mathcal{C}_d^{i,j}$ provides a fine-grained, contrastive evidence to distinguish class $y^i$ from a confusable class $y^j$.

In this way, the strong LLM generalization ability amends the data shortage problem by optimizing concept matching effectiveness, which further improves the prediction accuracy. From a causal perspective, our architecture forms a sufficient justification for each prediction by jointly answering "why is it class $y^i$?" via $\mathcal{C}_p$ and "why is it not the confusable class $y^j$?" via $\mathcal{C}_d$.

## 3.2 CONCEPT GENERATION

**Concept Proposal** $\mathcal{C}_p$ and $\mathcal{C}_d$ are extracted by distinct strategies to guarantee their effectiveness. $\mathcal{C}_p$ are designed to ensure high recall of correct candidate labels. Thus, we prompt the LLM with multiple samples from the same target class and instruct it to induce their shared, common features. $\mathcal{C}_d$, in contrast, must exhibit high inter-class separability to distinguish between similar candidates. To achieve this, we prompt the LLM with samples from two classes, $y^i$ and $y^j$, and instruct it to identify concepts that are characteristic of class $y^i$ but are rarely or never present in class $y^j$. Details of the concept proposal are shown in Appendix. A.3. The pair-wise $\mathcal{C}_d$ proposal process causes a quadratic cost $O(|\mathcal{Y}|^2)$. To mitigate such cost for large-scale scenarios ($|\mathcal{Y}| \gg 10$), one can cluster classes based on $\mathcal{C}_p$ similarity and restrict $\mathcal{C}_d$ generation to the top-$K$ semantic neighbors, thereby reducing complexity from to $O(|\mathcal{Y}|K)$, $K \ll |\mathcal{Y}|$.

In detail, all concepts have a `name` for identification and a human-understandable `description` for semantic matching. The `description` is crucial as it will be converted into a vector embedding for similarity calculation. Therefore, the prompt explicitly instructs the LLM to write a description that is comprehensive and semantically rich, ensuring an accurate representation for the subsequent matching process. Concrete examples are shown in Sec. 4.7.

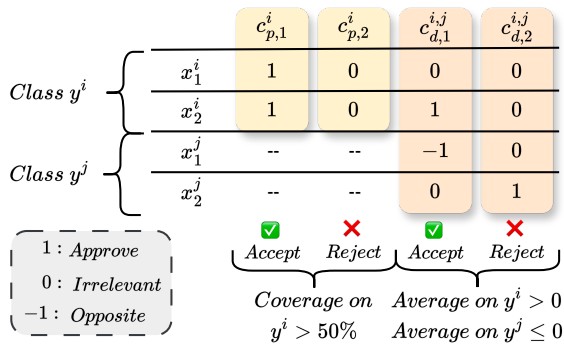

Figure 2: Illustrative example of LLM-based Concept Annotation. Rows represent text samples ($x_i$), columns represent candidate concepts ($c_j$). Values $\{-1, 0, 1\}$ denote semantic consistency.

**Concept Annotation and Selection** Following the generation of candidate concepts, we employ a logical annotation and selection process to prune low-quality concepts that fail their functionalities within the limited training data. This process starts with an annotation procedure. For each candidate prototypical concept $c_p^i$ and a training sample $x^i$ from $y^i$, we require an LLM to annotate their logical relationship with a score $a(x^i, c_p^i) \in \{-1, 0, 1\}$, which means contradiction, irrelevance, or logical consistency, respectively (see Figure 2). And for

each discriminative concept $c_d^{i,j}$, the same annotation procedure will be applied to samples from both related labels $y^i$ and $y^j$.

After annotation, we apply the following rules to select high-quality concepts:

- For each $c_p^i$, it is only selected into $\mathcal{C}_p^i$ when a certain threshold of samples from $y^i$ are consistent.

- For each $c_d^{i,j}$, acceptance into a pairwise set $\mathcal{C}_d^{i,j}$ demands satisfaction of possitive scores for samples from target class $y^i$ and non-positive scores on samples from contrastive class $y^j$.

concept generation halts when all support samples satisfy the activation criteria (i.e., $a(x^i, c) = 1$ for at least one $c_p^i$ and $a(x^i, c) = 0$ for all $c_d^{\cdot,i}$). To ensure the reliability of this automated process, we rigorously validated both the internal consistency of the LLM's scoring and its alignment with human judgment (see Sec. 4.6 and Appendix. A.7).

### 3.3 TWO-STAGE PREDICTION

Once the concept library is generated, StructCBM operates as a lightweight and LLM-free prediction engine. The prediction process follows a hierarchical decision-making procedure that is transparent and mirrors human reasoning: it first employs the prototypical concepts ($\mathcal{C}_p$) to identify a high-potential candidate set of labels ($\mathcal{Y}_k$), and then uses the discriminative concepts ($\mathcal{C}_d$) to confirm the final prediction through a fine-grained, contrastive analysis.

The prediction process requires the similarity comparison between concepts and samples, and we follow a common practice to use the text embedding similarities. With a text embedding model $e(\cdot)$ that encodes texts into a unified $d$-dimensional vector space, the semantic relevance between $x$ and $c$ is then quantified by the cosine similarity of their respective embeddings: $\text{sim}(x, c) = \cos(e(x), e(c))$. This design choice is crucial for efficient inference, as it bypasses the need for expensive LLM calls during the online prediction phase.

#### 3.3.1 STAGE 1: PROTOTYPICAL PRUNING

In this stage, our model aims to identify the most promising candidate labels by estimating the alignment of the input $x$ with each class's prototypes. For each class $y^i \in \mathcal{Y}$, we calculate a **Prototypical Support Score**, $s_p^i(x)$, by averaging the similarity of the top-$m$ most relevant prototypical concepts:

$$s_p^i(x) = \frac{1}{m} \sum_{c_p^i \in \text{top-}m(\mathcal{C}_p^i, x)} \text{sim}(x, c_p^i), \tag{2}$$

where top-$m(\mathcal{C}_p^i, x)$ is the set of $m$ prototypical concepts from $\mathcal{C}_p^i$ that have the highest similarity scores with sample $x$. We represent the prototype scores for all classes as a vector $\mathbf{s}_p(x) = [s_p^1(x), \ldots, s_p^n(x)]^\mathsf{T} \in \mathbb{R}^{|\mathcal{Y}|}$. We then select the candidate set $\mathcal{Y}_k$ by taking the top-$k$ classes with the highest $s_p^i(x)$ scores. This effectively prunes the label space, focusing subsequent analysis on the most plausible options.

#### 3.3.2 STAGE 2: DISCRIMINATIVE RE-RANKING

The second stage finalizes the prediction by performing a fine-grained, contrastive analysis exclusively among the candidate labels in $\mathcal{Y}_k$. This stage quantifies the unique discriminative evidence for each candidate by a **Relative Discrimination Score** $s_d^{i,j}$. For any pair of candidate labels $(y^i, y^j) \in \mathcal{Y}_k$, $s_d^{i,j}(x)$ quantifies the net evidence favoring $y^i$ over $y^j$ as:

$$s_d^{i,j}(x) = \max\{\text{sim}(x, c_d^{i,j}) | c_d^{i,j} \in \mathcal{C}_d^{i,j}\} - \max\{\text{sim}(x, c_d^{j,i}) | c_d^{j,i} \in \mathcal{C}_d^{j,i}\}. \tag{3}$$

We then aggregate these pairwise comparisons into a **Net Discrimination Score**, $s_d^i(x) = \sum_{y^j \in \mathcal{Y}_k, j \neq i} s_d^{i,j}(x)$, for each candidate $y^i$. The final prediction for sample $x$ is then determined by combining the initial prototype evidence with this fine-grained discriminative evidence. We compute a final score, $s^i(x)$, for each candidate $y^i \in \mathcal{Y}_k$:

$$s^i(x) = \alpha \cdot s_p^i(x) + (1 - \alpha) \cdot s_d^i(x), \tag{4}$$

where $\alpha \in [0, 1]$ is a balancing hyperparameter. The prediction will be made on the label with the highest $s^i(x)$, that is $\hat{y} = \arg\max_{y^i \in \mathcal{Y}_k} s^i(x)$.

## 3.4 Concept Tuning

Finally, the framework closes the learning cycle with a concept tuning stage. This closed-loop mechanism uses classification errors from the prediction module as feedback to the LLM, guiding it to improve the concept library in semantical and logical ways. For **Semantic Tuning**, the LLM rewrites existing concept descriptions to strengthen their similarities to same-class samples and fix the errors. For more persistent errors, **Logical Tuning** is triggered, where the misclassified samples serve as targeted input to complete the concepts by re-invoking the generation process from Sec. 3.2.

**Semantic Tuning**   This stage rewrites existing concept descriptions to better align their semantical meanings with their logical purpose. Given a residual misclassified sample $r$, there are two possible causes to the error, recall error and ranking error, which are correspondingly handled during this process. To prevent concept drift and overfitting to single error samples, we enforce a **regularization constraint**: any updated concept (whether $c_p$ or $c_d$) must maintain or improve its average similarity to the set of correctly matched positive samples while addressing the specific error. The algorithm details are shown in Algorithm 1, and related prompts are shown in Appendix. A.4.

For a recall error, where the correct label $y_{true}$ is not in the candidate set, we identify the relevant prototypical concept $c_p$ and prompt the LLM to generate paraphrases. The paraphrase that maximizes the average similarity with all of its supported samples is chosen to replace the original. For a ranking error, where a sample is misclassified, we identify the most misguiding discriminative concept $c_d$ and rewrite its description to minimize its similarity to the error-inducing sample.

**Logical Tuning**   In case there are hard samples that semantical tuning fails to adapt, we generate new concepts specially to cover these samples. The generation process for prototypical and discriminative concepts is the same in Sec. 3.2, with a special focus on hard samples.

---

**Algorithm 1** Semantic Tuning Function

1: **Input:** A residual case $r$, $\mathcal{C}$, positive examples $X_i$
2: **Output:** Updated $\mathcal{C}'$
3: **if** $type(r) = $ RecallError **then**
4:     Let $c_p^i$ be a relevant concept.
5:     $[c_p^{i'}]_{\text{cand}} \leftarrow \text{Prompt}(c_p^i, r, X_i)$
6:     $c_p^{i*} \leftarrow \arg\max_{c_p^{i'}} \sum_{x'|a(c_p^i, x')=1} \text{sim}(c_p^{i'}, x')$
7:     $\mathcal{C}^i \leftarrow \mathcal{C}^i \setminus \{c_p^i\} \cup \{c_p^{i*}\}$
8: **else if** $type(r) = $ RankingError **then**
9:     $c_d^{i,j} \leftarrow \arg\max_{c \in \mathcal{C}_d^{i,j}} \text{sim}(r, c)$
10:     $[c_d^{i,j'}]_{\text{cand}} \leftarrow \text{Prompt}(c_d^{i,j}, r)$
11:     $c_d^{i,j*} \leftarrow \arg\min_{c_d^{i,j'}} \text{sim}(c_d^{i,j'}, r)$
12:     **if** $\text{sim}(c_d^{i,j*}, X_i) > \text{sim}(c_d^{i,j}, X_i)$ **then**
13:         $\mathcal{C}^{i,j} \leftarrow \mathcal{C}^{i,j} \setminus \{c_d^{i,j}\} \cup \{c_d^{i,j*}\}$
14:     **end if**
15: **end if**
16: **return** Updated $\mathcal{C}$

---

## 3.5 Embedding Model Finetuning

At the end of concept library construction, we have an additional model training module that finetunes the embedding model to achieve a better matching performance between samples and concepts. The training data $\mathcal{P}$ is constructed by the samples and concepts that support the same label:

$$\mathcal{P} = \cup_{y^i \in \mathcal{Y}} \mathcal{P}^i, \mathcal{P}^i = \left\{ \{(x^i, c_p^i,)\}_{c_p^i \in \mathcal{C}_p^i} \cup \{(x^i, c_p^j)\}_{c_p^j \in \mathcal{C}_p^j} \right\}_{x^i \in \mathcal{X}^i}. \quad (5)$$

$\mathcal{C}_d$ are excluded from data synthesis for their lack of generalizable information. After the preparation of $\mathcal{P}$, we optimize the embedding model $e(\cdot)$ with cosine similarity loss:

$$\hat{e} = \arg\min_e \sum_{(x^i, c^j) \in \mathcal{P}} ||\delta(i, j) - \cos(e(x^i), e(c^j))||_2, \quad (6)$$

where $\delta(i, j) = 1$ if $i = j$, and otherwise 0. In such a way, we could improve the embedding similarity accuracy and make better predictions. Note that such a procedure does not hinder the interpretability of StructCBM, since the prediction process is still honestly reflected through concepts.

## 4 Experiments

We conduct extensive experiments on several text classification benchmarks with few-shot setting to evaluate the efficacy of our proposed framework, StructCBM. We introduce the experiment setup

Table 2: Main results on four benchmark datasets. Few-shot results are evaluated on a 10-shot per label setting. RoBERTa and DistilBERT are also finetuned on full training dataset to present a performance upper bound.

| | Model | SST2 | | AGNews | | MedAbs | | FinaQuery | |
|---|---|---|---|---|---|---|---|---|---|
| | | Acc | F1 | Acc | F1 | Acc | F1 | Acc | F1 |
| Black-box Full | RoBERTa | 0.9462 | 0.9456 | 0.9508 | 0.9460 | 0.6457 | 0.6394 | 0.8561 | 0.8131 |
| | DistilBert | 0.8979 | 0.8978 | 0.9426 | 0.9428 | 0.6291 | 0.6296 | 0.8411 | 0.7600 |
| Black-box Few-shot | RoBERTa | 0.5008 | 0.3336 | 0.6284 | 0.7475 | 0.3778 | 0.5917 | 0.5056 | 0.6672 |
| | DistilBert | 0.5859 | 0.5728 | 0.8359 | 0.8374 | 0.4349 | 0.4280 | 0.5289 | 0.4842 |
| | Deepseek-Direct | 0.9610 | 0.9614 | 0.8047 | 0.7911 | 0.6395 | 0.6100 | 0.8499 | 0.8510 |
| | Deepseek-ICL | 0.9630 | 0.9630 | 0.7900 | 0.7705 | 0.6374 | 0.6060 | 0.8360 | 0.8389 |
| White-box Few-shot | TBM | 0.4526 | 0.4431 | 0.2490 | 0.2439 | 0.2890 | 0.2852 | 0.2788 | 0.2684 |
| | CBLLM | 0.6326 | 0.6310 | 0.6834 | 0.6853 | 0.3521 | 0.3478 | 0.3465 | 0.3252 |
| | C$^3$M | 0.5392 | 0.5341 | 0.5568 | 0.6682 | 0.3109 | 0.5671 | 0.5056 | 0.6672 |
| | SparseCBM | 0.5727 | 0.5628 | 0.6284 | 0.7475 | 0.3778 | 0.5917 | 0.4427 | 0.6656 |
| | StructCBM (Ours) | 0.8390 | 0.8371 | 0.8545 | 0.8539 | 0.6070 | 0.6042 | 0.7742 | 0.7763 |

in Sec. 4.1, including the datasets, baselines, and evaluation metrics. We compare the effectiveness of StructCBM with baselines in Sec. 4.2, evaluate the contribution of each component in Sec. 4.3, examine the robustness of StructCBM in Sec. 4.4, and analyze its scalability in Sec. 4.5. Besides, we quantitatively and qualitatively evaluate the interpretability of StructCBM in Sec. 4.6 and Sec. 4.7, respectively.

## 4.1 EXPERIMENT SETUP

**Datasets.** We include four public text-classification datasets from diverse domains to ensure a comprehensive evaluation: **AGNews** on news topics (Zhang et al., 2015), **SST2** on sentiment polarity (Socher et al., 2013), **MedAbs** on medical abstract subjects (Schopf et al., 2023), and **FinaQuery** on financial query category (Theerthala, 2025). For each of these datasets, we create a **10-shot** training split by randomly sampling 10 examples per class and keep the original test dataset for evaluation. As there is no test set for FinaQuery, 25% of the full data is randomly sampled as test data. More details are shown in Appendix. A.5.

**Implementation Details.** We extract, annotate, and logically refine both types of concepts by calling API from DeepSeek-V3 (DeepSeek, 2025), where the temperature is set to 0.7 and context length set to 8792. To align with baselines (Sun et al., 2025), the text embedding model is `all-mpnet-base-v2` (Song et al., 2020) with implementation from the sentence transformer package. The hyperparameters we used for inference are attached in Appendix. A.6.

**Baselines.** To comprehensively evaluate StructCBM, we benchmark our model against three categories of state-of-the-art baselines. First, we compare with leading LLM-based CBMs, including **TBM** (Ludan et al., 2024), **C$^3$M** (Tan et al., 2024b), **SparseCBM** (Tan et al., 2024a), and **CB-LLMs** (Sun et al., 2025). Second, we evaluate small black-box models, specifically **RoBERTa-base** (Liu et al., 2019) and **DistilBERT** (Sanh et al., 2020), under both few-shot and fully supervised settings. Finally, to establish a high-performance upper bound, we benchmark against **DeepSeek V3** (DeepSeek, 2025) using both few-shot in-context learning and direct inference.

**Evaluation Metric.** Model performance is assessed using standard classification metrics, including **Accuracy** and **Macro-F1 score**. Accuracy provides an overall measure of correct predictions, while Macro-F1 is particularly useful in multi-class or imbalanced scenarios, offering a balanced view of precision and recall across all classes.

## 4.2 MAIN RESULTS AND ANALYSIS

We compare StructCBM against both white-box and black-box baselines under a strict 10-shot per label setting. For a performance upper bound, we also include RoBERTa and DistilBert models fine-tuned on the full training set. The comprehensive results are presented in Table 2. StructCBM establishes a new state-of-the-art for interpretable few-shot learning by significantly outperforming

Table 3: Ablation study on different components of StructCBM. The results reported are based on a concept weighting of $\alpha = 0.75$. "Reg." denotes the stricter regularization constraint during tuning. The best results for each dataset are shown in **bold**.

| Components | | | | | SST2 | AGNews | MedAbs | FinaQuery |
|:---:|:---:|:---:|:---:|:---:|:---:|:---:|:---:|:---:|
| $\mathcal{C}_p$ | $\mathcal{C}_d$ | Refine | Reg. | Train | | | | |
| ✓ | | | | | 0.7507 | 0.7224 | 0.5530 | 0.6051 |
| ✓ | ✓ | | | | 0.8029 | 0.7930 | 0.5547 | 0.6491 |
| ✓ | ✓ | ✓ | | | 0.7677 | 0.8141 | 0.5900 | 0.6900 |
| ✓ | ✓ | ✓ | ✓ | | 0.8127 | 0.8235 | 0.6004 | 0.7177 |
| ✓ | ✓ | ✓ | ✓ | ✓ | **0.8390** | **0.8545** | **0.6070** | **0.7742** |

prior white-box CBMs across all datasets. This result underscores the limitations of existing CBMs in data-scarce scenarios and validates the effectiveness of our architecture. Furthermore, StructCBM demonstrates strong competitiveness against powerful black-box LLMs, especially on semantically rich datasets like AGNews and MedAbs. Notably, on AGNews, our model surpasses the 10-shot Deepseek-ICL, and on MedAbs, our performance is comparable to the zero-shot Deepseek-Direct, all while offering the crucial advantages of full interpretability. A performance gap remains on datasets with less semantic density, such as SST2 and FinaQuery. We attribute this to the advantage of the vast world knowledge embedded in black-box models for these specific tasks.

## 4.3 ABLATION STUDY

To validate the effectiveness of each key component in StructCBM, we conduct a comprehensive ablation study, with results presented in Table 3. For these experiments, the concept weighting parameter is set to $\alpha = 0.75$. Our analysis focuses on two core aspects of our design: the dual-layer concept structure and the iterative refinement process.

First, incorporating discriminative concepts ($\mathcal{C}_p + \mathcal{C}_d$) consistently improves performance over using only prototypical concepts ($\mathcal{C}_p$) across all datasets, confirming that fine-grained boundary analysis is crucial for resolving ambiguity. Next, we evaluate the refinement (Refine) and regularization (Reg.) stages. While unconstrained refinement boosts performance on semantically rich datasets (e.g., AGNews), it causes overfitting on the binary SST-2 dataset (accuracy drops from 0.8029 to 0.7677). Crucially, introducing the regularization constraint (Reg.) successfully resolves this instability, recovering SST-2 accuracy to 0.8127. Finally, the embedding model training (Train) further enhances performance across all benchmarks, demonstrating the benefits of aligning the representation space.

## 4.4 SENSITIVITY AND ROBUSTNESS ANALYSIS

**Internal Concept Weighting ($\alpha$).** We vary the weight $\alpha \in [0, 1]$ to balance prototypical ($\mathcal{C}_p$) and discriminative ($\mathcal{C}_d$) concepts. As shown in Figure 3, performance peaks when $\alpha \geq 0.5$, favoring $\mathcal{C}_p$. **We attribute this to their functional roles:** $\mathcal{C}_p$ captures the core, generalized semantics (high-recall signal), ensuring predictions land in the correct semantic vicinity. In contrast, $\mathcal{C}_d$ focuses on fine-grained boundary distinctions (high-precision signal). In few-shot settings, the robust signal from $\mathcal{C}_p$ serves as the primary driver, while $\mathcal{C}_d$ acts as a critical refiner for hard negatives.

Figure 3: Sensitivity of weighting $\alpha$. $\alpha \geq 0.5$ yields optimal results.

**External Robustness.** As detailed in Table 4, we further validated robustness against three components: (1) **LLMs:** Performance remains robust across proprietary (GPT-4o (OpenAI, 2023)) and open-weights models (DeepSeek, Qwen3-Plus (Yang et al., 2025)). (2) **Embedding Models:** StructCBM effectively leverages stronger backbones. Switching from *MiniLM* (Wang et al., 2020) to *Qwen3-embedding-0.6B* (Zhang et al., 2025) (noted as Qwen-emb) boosts accuracy (e.g., +5.68% on AGNews), confirming our framework scales with better representations. (3) **Data Scarcity:** Consistent gains are observed from 1-shot to 10-shot. Notably, the framework achieves respectable performance even in the extreme 1-shot setting (e.g., 0.6624 on AGNews), showing strong few-shot resilience.

Table 4: Robustness analysis across different LLMs (left), embedding models (middle), and few-shot settings (right) on accuracy. StructCBM shows consistent stability across all configurations.

| (a) Different LLM Generators | | | | (b) Embedding Backbones | | | | (c) Few-shot Settings | | | |
|---|---|---|---|---|---|---|---|---|---|---|---|
| Dataset | DeepSeek | GPT-4o | Qwen | Dataset | MiniLM | mpnet | Qwen-emb | Dataset | 1-shot | 5-shot | 10-shot |
| SST-2 | 0.8390 | **0.8680** | 0.8011 | AGNews | 0.8121 | 0.8545 | **0.8689** | AGNews | 0.6624 | 0.8472 | **0.8545** |
| AGNews | 0.8545 | **0.8567** | 0.8456 | MedAbs | 0.5907 | 0.6070 | **0.6143** | FinQuery | 0.5873 | 0.7360 | **0.7742** |

## 4.5 EFFICIENCY AND SCALABILITY ANALYSIS

The efficiency of StructCBM is assessed by distinguishing between the one-time offline construction phase and the online inference phase. The primary computational overhead arises *exclusively* during offline construction. Crucially, once the concept library is built, the **online inference phase** is highly efficient, incurring zero marginal API costs (see Table 5). Furthermore, the memory footprint for concept embeddings is negligible ($<$10MB) compared to the backbone.

Table 5: Efficiency comparison. StructCBM incurs API costs *only once* during construction, whereas LLMs incur recurring costs per inference.

| **Dataset** | **Method** | **API Calls** | **Type** |
|---|---|---|---|
| SST-2 | DeepSeek | 1,820 | Recurring |
| | **Ours** | **75** | **One-time** |
| AGNews | DeepSeek | 7,600 | Recurring |
| | **Ours** | **173** | **One-time** |

Regarding offline construction cost, our empirical analysis indicates that the average API consumption per unit (class or pair) remains consistent across datasets. This stability is driven by our explicit stopping criteria (e.g., ensuring sufficient coverage for $\mathcal{C}_p$ and validity for $\mathcal{C}_d$), which effectively bound the generation process. We provide a detailed quantitative breakdown of these unit costs in Appendix A.8.1. Although unit costs are stable, the number of pairwise comparisons grows quadratically with classes. To ensure scalability for scenarios with hundreds of labels ($|\mathcal{Y}| \gg 10$), we propose a similarity-based pre-filtering strategy (see Appendix A.8.1) that restricts discriminative concept generation to semantic neighbors, reducing complexity from $O(|\mathcal{Y}|^2)$ to $O(|\mathcal{Y}|K)$.

## 4.6 INTERPRETABILITY AND RELIABILITY ANALYSIS

To rigorously validate interpretability, we recruited three evaluators with Finance backgrounds to assess a stratified sample set. As shown in Table 6, evaluators rated the concepts as providing highly sufficient evidence for predictions (**4.39/5.0**). Furthermore, a blind agreement test demonstrated that the LLM's automated annotation matched human expert consensus in **96.43%** of cases, confirming the trustworthiness of our automated pipeline. Beyond human alignment, we verified internal stability on the MedAbs dataset, where the LLM maintained **98.6% consistency** across independent trials, proving that our framework produces deterministic features rather than stochastic hallucinations. Details are presented in Appendix. A.7

Table 6: Quantitative validation of concept interpretability and reliability. Human evaluation (Expert) confirms semantic validity, while LLM robustness tests confirm stability.

| **Evaluation Metric** | **Score** |
|---|---|
| *Human Evaluation* | |
| Contribution Faithfulness (1-5) | **4.39** |
| Annotation Agreement (%) | **96.43** |
| *Annotation Consistency* | |
| MedAbs (%) | **98.60** |

## 4.7 CASE STUDY: TWO-STAGE REASONING IN ACTION

To intuitively demonstrate our model's reasoning process, we present a case study MedAbs about a vascular surgical procedure. As illustrated in Figure 4, our model follows a two-stage process to arrive at the correct classification. In Stage 1, the prototype score for each potential label is calculated by averaging the similarity scores of its most relevant $\mathcal{C}_p$. For clarity, our figure displays only the most similar prototypical concept of the top-2 candidate labels. The broad and general nature of the *General pathological conditions* label, led by its top concept 'Therapeutic intervention management', resulted in a high $s_p$. Despite this, our model successfully recalled the true label, *Cardiovascular diseases* into the top-2 candidate set.

**Text:** Retrograde recanalization of an occluded posterior tibial artery by using a posterior tibial cutdown: two case reports. Recanalization of two occluded posterior tibial arteries was successfully achieved by utilizing a retrograde approach via a posterior tibial artery cutdown at the level of the ankle. Both cases were previously unsuccessfully attempted by using an antegrade approach. Thus, the choice of access vessel (arterial entry site) becomes a crucial determinant of angioplasty success.

**Stage-one prediction**: *Label 1* General pathological conditions, *Label 2* Cardiovascular diseases (True label)

**Concept type:** *Prototype*
**Concept name:** Therapeutic intervention management and evaluation
**Description:** Analysis of treatment modalities encompassing procedural implementation, technical approaches, comparative effectiveness of interventions, and evaluation of therapeutic results and patient outcomes.
**Support label:** *Label 1* General pathological conditions

**Concept type:** *Prototype*
**Concept name:** reperfusion intervention outcomes
**Description:** Clinical results and efficacy measures following procedures aimed at restoring blood flow to ischemic cardiac tissue, including success rates, complications, and functional improvements
**Support label:** *Label 2* Cardiovascular diseases

$$\text{Prototype Support Score } s_p^1 = 0.3501$$

$$\text{Prototype Support Score } s_p^2 = 0.3341$$

**Stage-two prediction**: *Label 2* Cardiovascular diseases ✅

**Concept type:** *Discriminative*
**Concept name:** non-vascular anatomical structure repair
**Description:** References to surgical repair or reconstruction of non-vascular anatomical structures such as urethral defects, spermatic cord, ocular implants, or anal fistulas using techniques like onlay coverage or specific implant management
**Support label:** *Label 1* General pathological conditions

**exclude_label:** *Label 2* Cardiovascular diseases

**Concept type:** *Discriminative*
**Concept name:** vascular surgical intervention
**Description:** Mentions of surgical procedures specifically targeting blood vessels, arteries, or veins for repair, bypass, or revascularization purposes, including procedures like infrainguinal bypass, femoropopliteal bypass, femorodistal bypass, or arterial graft placement
**Support label:** *Label 2* Cardiovascular diseases

**exclude_label:** *Label 1* General pathological conditions

$$\text{Relative Discriminative Score } s_d^{1,2} = -0.1241$$

$$\text{Relative Discriminative Score } s_d^{2,1} = 0.1241$$

$$\text{Final Score}: \alpha s_p^1 + (1-\alpha)s_d^{1,2} = 0.2078$$

$$\text{Final Score}: \alpha s_p^2 + (1-\alpha)s_d^{2,1} = 0.2711$$

Figure 4: An illustrative example of our model's two-stage prediction process. **Stage 1:** A prototype score, calculated by averaging the similarities of the top three concepts for each label, is used to select a candidate set. For clarity, the figure displays the single most influential prototypical concept for each of the top two candidates. **Stage 2:** Discriminative concepts provide decisive contrastive evidence, correcting the final ranking in favor of the true label.

In Stage 2, the model leverages discriminative concepts to resolve the ambiguity. The concept *vascular surgical intervention* perfectly matches the text, providing positive evidence for the true label. Conversely, the concept *non-vascular anatomical structure repair* provides crucial negative evidence against the competing general label. This contrastive scoring corrects the initial ranking, leading to the accurate final prediction of *Cardiovascular diseases* with a higher final score. This case clearly shows how our architecture first ensures high recall and then uses discriminative evidence for precise, interpretable classification.

## 5 LIMITATIONS AND FUTURE WORK

While our framework demonstrates strong performance, we identify two key directions for future research. First, the pairwise formulation of our discriminative concepts $\mathcal{C}_d$ scales quadratically with the number of classes, posing a challenge for tasks with large label sets. Future work could address this by exploring on-demand concept generation for only the most confusable class pairs (also discussed in Appendix. A.8). Second, from a causal perspective, our reasoning process provides a sufficient condition for its predictions, but not a necessary one. A key future direction is to leverage the counterfactual generation capabilities of LLMs to compensate for necessity.

## 6 CONCLUSION

In this work, we introduced StructCBM, a novel framework for few-shot text classification that successfully merges the performance of large language models with the efficiency and interpretability of Concept Bottleneck Models. Our approach leverages a dual-level, prototypical-discriminative concept architecture and an error-driven refinement loop to generate high-quality concepts from limited data, enabling a lightweight, LLM-free inference process based on direct concept similarity. Extensive experiments confirm that StructCBM's performance is comparable to strong black-box baselines and surpasses prior CBMs, establishing a new paradigm for trustworthy few-shot learning.

ACKNOWLEDGMENTS

This work was supported by the General Research Fund (GRF) of the Hong Kong Research Grants Council (RGC) under grants 16310222, 16308221, and 16308421.

ETHICS STATEMENT

This work adheres to the ICLR Code of Ethics. In this study, no human subjects or animal experiments was involved. All datasets used were sourced in compliance with relevant usage guidelines, ensuring no violation of privacy. We have taken care to avoid any biases or discriminatory outcomes in our research process. No personally identifiable information was used, and no experiments were conducted that could raise privacy or security concerns. We are committed to maintaining transparency and integrity throughout the research process.

REPRODUCIBILITY STATEMENT

The code and datasets for this paper are publicly available at `https://github.com/alexiszlf/StructCBM`. The implementation details are shown in Sec. 4.1 and Appendix. A.6.

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

# A APPENDIX

## A.1 USE OF LARGE LANGUAGE MODELS

Large language models are used for finding related works and polishing the paper presentation. The authors manually surveyed representative related works and used LLM only for supplementary paper discovery. All the related works returned from LLMs are checked and reviewed by at least one author, and all of the references are guaranteed authentic and real. The content that is polished by LLMs is always based on the original draft written by the authors, which is also carefully reviewed by at least one of the authors.

## A.2 ILLUSTRATIVE COMPARISON BETWEEN DIFFERENT TEXT CLASSIFICATION METHODS

We compare the pipeline between direct LLM inference, other CBM models and our models under text classification scenarios in Figure 5 below.

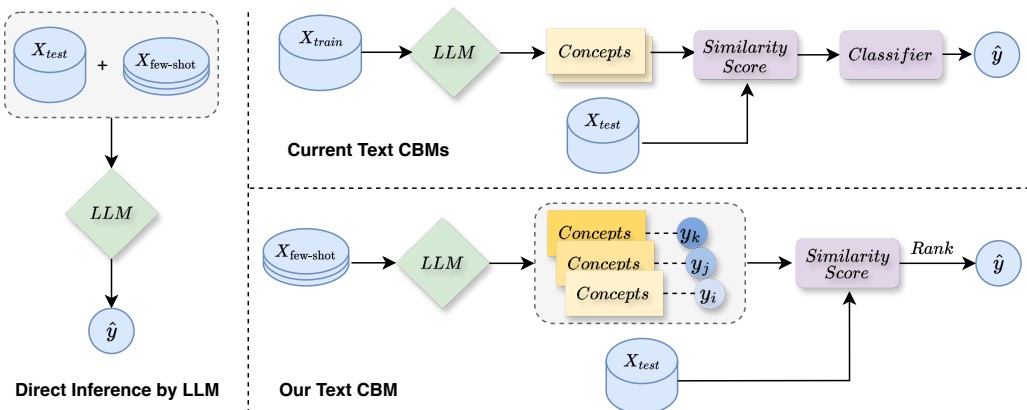

Figure 5: A comparison of conceptual workflows. **Top:** Existing LLM-based CBMs typically require a large training set ($X_{\text{train}}$) to generate concepts and then train a separate classifier to map concept scores to labels. **Bottom:** Our proposed method utilizes a few-shot set ($X_{\text{shot}}$) to generate concepts with a built-in one-to-one correspondence to labels, enabling direct prediction via similarity ranking without a trainable classifier.

## A.3 PROMPT DESIGN FOR CONCEPT GENERATION

In this section, we provide a detailed overview of the prompt structures used to guide the Large Language Model (LLM) in generating our dual-layer concept library. The design of these prompts is crucial for eliciting high-quality, specialized concepts tailored for our few-shot learning framework.

A key instruction embedded in both prompt types is the explicit guidance that the generated concept "description" will be subsequently used for **embedding similarity matching**. This encourages the LLM to produce descriptions that are not only semantically accurate but also rich in contextual cues and varied expressions, making them more robust for downstream similarity calculations.

Furthermore, given the few-shot nature of our task, we cannot rely on statistical metrics to guide the concept generation process. To mitigate the risk of generating redundant or overlapping concepts, we

implement a simple yet effective de-duplication strategy. In each generation request, we explicitly provide the LLM with a list of **previously generated concepts** for that class or class pair, and instruct it to avoid repetition, as shown in both Figure 6 and Figure 7.

```
You are a {DATASET_CONFIGS[dataset_name]['domain']} reasoning expert.
We are developing a concept-based classifier for label: "{label}".
Below are {len(samples)} samples that are confirmed to belong to label: "{label}":
{sample_texts}
Your task is to identify 1-2 semantic concepts that help determine whether
a new sample should belong to this label.
We have already identified the following known concepts (please do not repeat these): {known_list}
Propose 1-2 new **high-level concepts** that help identify samples belonging to label "{label}".
Each concept should:
- Capture an abstract, broadly applicable semantic pattern commonly observed in this label
- Be general enough to match varied expressions across samples
- Avoid overfitting to any single phrase or example
⚠ Important:
The "description" will later be converted into an embedding for similarity matching.
Therefore:
- Write in a clear, professional, academic style
- Naturally include key semantic cues and related expressions that frequently co-occur with this label
- Ensure the description is comprehensive enough to cover multiple wording variations
For each concept, return:
1. `name`: a short abstract definition of the concept (this will be used as the concept name)
2. `description`: a generalized statement that can match other samples semantically
3. `explanation`: reasoning why this concept is useful for classification
Strictly return a JSON list like this:
```json
[{DATASET_CONFIGS[dataset_name]['example_propose_concept']}]
```

Figure 6: The prompt structure for generating prototypical concepts ($\mathcal{C}_p$). The LLM is tasked with identifying high-level, generalizable semantic patterns from a few examples of a single target label. It is explicitly instructed not to repeat concepts from the provided "known_list".

### A.4 PROMPT DESIGN FOR SEMANTIC TUNING

The prompts for our Semantic Tuning stage are tailored to the distinct roles of prototypical ($\mathcal{C}_p$) and discriminative ($\mathcal{C}_d$) concepts.

To maintain the generality of a prototypical concept during rewriting, our prompt includes not only the misclassified sample but also a reference set of correctly classified examples, as shown in Figure 8. This crucial step guides the LLM to capture a broader semantic essence and avoid overfitting to a single error case.

In contrast, rewriting a Discriminative Concept aims to sharpen its specificity. The prompt, illustrated in Figure 9, focuses solely on the misclassified sample and the misguiding concept. It instructs the LLM to revise the description to be more semantically dissimilar to the error, thereby directly addressing the failure of discrimination.

### A.5 DATASET DETAILS

The benchmark datasets are public text-classification datasets. Their statistical information is listed in Table. 7.

### A.6 HYPERPARAMETER DETAILS

The weight $\alpha$ that determines the contribution of $\mathcal{C}_p$ and $\mathcal{C}_d$ is selected from $\{0, 0.25, 0.5, 0.75, 1\}$, and the detailed effect analysis of the performances between different choices and their effects are

```
You are a {DATASET_CONFIGS[dataset_name]['domain']} reasoning expert.
We are building a concept-based classifier to
**support label `{label_i}`** and **reject label `{label_j}`**.
Your task is to discover **exclusive concepts** that help distinguish
the target label "{label_i}" from a contrasting label "{label_j}".
To do this, analyze the examples from both labels provided below.
You should propose **1-2 concepts** such that:
- These concepts clearly appear or are implied in samples from label "{label_i}"
- But they are **strongly absent or even contradictory** in samples from label "{label_j}"
- If this concept appears in a sample, the sample **should not belong to label "{label_j}"**
For each concept:
1. Write a concise **concept definition** (e.g., "bilateral neurological symptoms")
2. Provide a **generalizable description** for semantic comparison
3. Justify why this concept excludes label "{label_j}" — using prior knowledge and the examples
These concepts will be used to **eliminate label `{label_j}`** when the concept is found.
⚠ Existing concepts already proposed (skip duplicates):
{existing}
Input examples:
{examples_text}
For each concept, return a JSON object with:
- `"name"`: short definition of the concept (e.g., `"surgical reintervention"`)
- `"description"`: a general description of how this concept appears in text
- `"explanation"`: why this concept helps distinguish label `{label_i}` from `{label_j}`
Return a JSON list like:
```json
[{DATASET_CONFIGS[dataset_name]['example_propose_cs_concept']}, ...]
```

Figure 7: The prompt structure for generating Discriminative Concepts ($\mathcal{C}_d$). The LLM is given examples from two confusable labels and tasked with discovering exclusive concepts that are present in the target label but strongly absent or contradictory in the contrasting label. The list of "existing" concepts is provided to prevent duplicates.

Table 7: Summary of Datasets Used

| Dataset | Train | Test | Classes | Task |
|---|---|---|---|---|
| SST2 (Socher et al., 2013) | 6.92k | 1.82k | 2 | Sentiment classification |
| AGNews (Zhang et al., 2015) | 120.00k | 7.60k | 4 | News topic classification |
| MedAbs (Schopf et al., 2023) | 11.6k | 2.89k | 5 | Patient condition classification |
| FinaQuery (Theerthala, 2025) | 7.04k | - | 8 | Query topic classification |

shown in Sec. 4.4. As for training the embedding model, the hyperparameters are set to 3 epochs, 4 samples per batch for MedAbs and 32 samples for other datasets, 0.00001 learning rates. All the other hyperparameters are the same as the default setting.

## A.7 HUMAN EVALUATION ON CONCEPT RELIABILITY AND VALIDITY

### A.7.1 EXPERIMENTAL SETUP

**Stratified Sampling Strategy.** To ensure diversity and representative coverage across decision boundaries, we employed a stratified sampling strategy. We randomly selected 2 samples from each class across three datasets (SST-2, AGNews, and FinancialQueries), resulting in a total of 28 evaluation instances ($N = 28$).

**Expert Evaluator Recruitment.** Given the domain-specific complexity of the *FinancialQueries* dataset, general crowdsourcing platforms (e.g., MTurk) may lack the necessary precision for evaluating financial concepts. To ensure high-quality, expert-level judgment, we recruited **three inde-**

```
You are a {DATASET_CONFIGS[dataset_name]['domain']} expert helping refine **concept descriptions**
for a {DATASET_CONFIGS[dataset_name]['domain']} classifier.
**Context:**
We have a concept factor "{factor_name}" under label "{label}" with the current description:
"{original_description}"
**Problem:**
Some samples that should match this concept are currently being misclassified (not recognized as belonging to "{label}").
**Samples that should match this concept but are currently misclassified:**
{misclassified_block}
**Samples that correctly match this concept:**
{correct_block}
**Your task:**
Revise the concept description to make it more likely to match the misclassified samples
while still being compatible with the correct samples.
**Requirements:**
1. **Focus on the misclassified samples** - what semantic patterns do they share that the current description might be missing?
2. **Maintain compatibility** with correct samples - don't break what's already working
3. **Avoid overfitting** - don't use exact phrases from the misclassified samples, but capture their semantic essence
4. **Keep it general** - the description should work for similar cases beyond these specific examples
5. **Use medical abstract language** - professional, precise, but not overly technical
**Output:**
Generate 5 candidate revised descriptions. Each should be a concise,
professional description that better captures the concept while remaining general.
Return strictly in this JSON format:
```json
{{
    "factor_name": "{factor_name}",
    "original_description": "{original_description}",
    "candidate_descriptions": [
        "candidate description 1",
        "candidate description 2",
        "candidate description 3",
        "candidate description 4",
        "candidate description 5"
    ]
}}
```
```

Figure 8: Prompt for rewriting a prototypical concept ($\mathcal{C}_p$). A reference set of correct samples ('correct_block') is provided to maintain generality and prevent overfitting.

**pendent evaluators with academic backgrounds in Finance**. Each sample was independently assessed by all three evaluators.

### A.7.2 TASK 1: CONTRIBUTION FAITHFULNESS

**Protocol.** The goal of this task is to assess the semantic validity of the concepts used for prediction. For each instance, evaluators were presented with:

1. The input text sample (drawn from the test set).
2. The model's predicted label.
3. The **Top-3 concepts** utilized by StructCBM for that prediction.

Evaluators were asked to rate, on a 5-point Likert scale, whether these concepts provided sufficient evidence to justify the classification (1 = Irrelevant/No Evidence, 5 = Perfect Evidence).

**Results.** StructCBM achieved a high average rating of **4.39 / 5.0**. This result strongly indicates that the concepts identified by our framework are not merely statistical artifacts but offer semantically meaningful and sufficient explanations that align closely with human expert reasoning.

### A.7.3 TASK 2: ANNOTATION RELIABILITY

**Protocol.** This task validates the reliability of the automated "Concept Annotation" phase (described in Section 3.2), where the LLM determines if a concept is present in a sample. We conducted a **blind agreement test** where evaluators manually determined the presence of specific concepts in

```
You are a {DATASET_CONFIGS[dataset_name]['domain']} reasoning assistant helping refine **contrastive concepts (Cs)** for classification.

A test sample was misclassified:
- True label: "{label_i}"
- Predicted label: "{label_j}"

📝 Misclassified sample:
{textwrap.indent(error_text.strip(), "  ")}
Candidate concepts involved:
✅ Concept from label {label_i} (should support this case):
- Name: {cs_ij_name}
- Description: {cs_ij_desc}
❌ Concept from label {label_j} (was wrongly matched):
- Name: {cs_ji_name}
- Description: {cs_ji_desc}
📖 Reference examples to guide generalization:
✔ Supporting examples (true {label_i}, match {cs_ij_name}):
{textwrap.indent(chr(10).join("- " + s for s in support_sample_texts), "  ") if support_sample_texts else "  (None)"}
✖ Excluding examples (true {label_j}, match {cs_ji_name}):
{textwrap.indent(chr(10).join("- " + s for s in exclude_sample_texts), "  ") if exclude_sample_texts else "  (None)"}
🔎 Important instructions:
1. We use **sentence embedding similarity** to decide whether a sample matches a concept.
- Therefore, revise the description of **{cs_ij_name}** so its embedding is **closer** to the misclassified sample (while still general enough).
- Revise the description of **{cs_ji_name}** so its embedding is **further away** from the misclassified sample.
2. Use the provided reference examples to keep the revised descriptions **general and robust**.
- Do NOT overfit to the exact misclassified sample wording.
- The revised description must still work well for the other examples.
3. Generate **2-3 candidate revised descriptions** for each concept.
- We will later select the best candidate.
- If a concept already works well, return it with `"candidates": []`.
---
📌 Your output: strictly JSON in this format:
```json
[
{{
    "type": "revised",
    "name": "{cs_ij_name}",
    "old_description": "{cs_ij_desc}",
    "candidates": ["...", "..."]
}},
{{
    "type": "revised",
    "name": "{cs_ji_name}",
    "old_description": "{cs_ji_desc}",
    "candidates": ["...", "..."]
}}
]
```

Figure 9: Prompt for rewriting a Discriminative Concept ($\mathcal{C}_d$). The goal is to enhance contrast by making the description more dissimilar to the misclassified sample.

the text (Yes/No) to establish a "Human Ground Truth." These manual annotations were then compared against the LLM's automated scoring.

**Results.** The LLM's automated annotation matched the human expert consensus in **27 out of 28 cases**, achieving an agreement rate of **96.43%**.

This high level of concordance empirically validates our reliance on LLMs for feature annotation. It demonstrates that, within the StructCBM framework, the LLM operates with a precision comparable to human domain experts, ensuring that the input features for the concept bottleneck are robust and trustworthy.

## A.8 DETAILED COST AND SCALABILITY ANALYSIS

To provide a comprehensive assessment of the computational overhead and scalability of StructCBM, we quantified the API consumption required for concept construction across different datasets. We further detail the scalability strategies proposed for large-scale scenarios.

A.8.1 QUANTITATIVE COST BREAKDOWN

Our empirical analysis indicates that while total costs naturally increase with the number of classes, the **unit cost per class (or class pair)** remains consistent.

**Understanding Unit Costs.** The reported unit costs encompass the full lifecycle of concept construction: **(1) Generation**, **(2) Annotation (Validation)**, and **(3) Retries**. The number of API calls is primarily determined by the strictness of our quality-control criteria. For example, in our 10-shot setting, we enforce a minimum requirement of 5 valid $c_p$ concepts per class and ensure that every sample is covered by at least one $c_p$ concept. The LLM typically satisfies these criteria within $\sim 7$ generation attempts (each followed by annotation), resulting in the observed average of $\sim 15$ calls for $\mathcal{C}_p$. This mechanism prioritizes concept quality over minimal API usage, ensuring robustness.

**Total vs. Unit Costs.** Table 8 presents the total API calls required for the 10-shot construction phase across three benchmarks. Despite the variation in total calls (from 75 to 1072), Table 9 confirms that the average consumption per unit is stable. For instance, the generation of discriminative concepts stabilizes at approximately 9 calls per pair in the 10-shot setting, regardless of the dataset size.

Table 8: Total API Calls for Concept Construction (10-shot setting).

| Dataset | Classes ($N$) | Total Pairs ($N^2 - N$) | Total API Calls |
|---|---|---|---|
| SST-2 | 2 | 2 | 75 |
| AGNews | 4 | 12 | 173 |
| FinaQuery | 8 | 56 | 1072 |

Table 9: Average API Calls Per Unit Breakdown. Note that $\mathcal{C}_p$ costs are per class, while $\mathcal{C}_d$ costs are per pair.

| Component | 1-shot | 5-shot | 10-shot |
|---|---|---|---|
| $\mathcal{C}_p$ Generation (per Class) | 4.83 | 11.12 | 15.38 |
| $\mathcal{C}_d$ Generation (per Pair) | 4.07 | 9.38 | 9.08 |
| Concept Tuning (per Class) | 0.00 | 9.77 | 13.01 |

**Cost Analysis of the Tuning Phase.** We observe that the tuning cost for *FinaQuery* is relatively higher ($\sim 1072$ total calls). This is primarily driven by the Concept Tuning phase, which is inherently dynamic: API usage correlates with the number of misclassified samples in the few-shot support set. However, this cost is fully controllable. In our experiments, we prioritized performance, allowing the tuning loop to run until convergence (or up to a high maximum round limit) to correct every possible error. In practice, this creates a flexible **Performance-Cost Trade-off**:

1) **Performance Mode (Used in Paper):** Allow extensive retries (e.g., 5+ rounds) to maximize few-shot accuracy, resulting in higher API calls.

2) **Efficiency Mode:** Restrict tuning to fewer rounds (e.g., 2 rounds of Semantic + 1 round of Logic Tuning). Empirical observations suggest that the majority of significant corrections occur in the first 2 rounds, allowing users to cap costs significantly with minimal impact on final accuracy.

A.8.2 EFFICIENCY COMPARISON: ONE-TIME VS. RECURRING COSTS

A critical advantage of StructCBM is that API costs are incurred **only once** during the offline construction phase. In contrast, LLM-based baselines (e.g., DeepSeek-ICL) require API calls for *every* inference sample.

As shown in Table 10, while the offline construction cost for StructCBM on AGNews is $\sim 173$ calls, the LLM baseline requires 7,600 calls to process the test set. Consequently, our method's cost does not scale with the volume of test data, ensuring superior scalability for large-scale deployment.

Table 10: Efficiency Comparison with LLM Baselines. StructCBM incurs a one-time construction cost, whereas LLMs incur recurring inference costs.

| Dataset | Method | API Calls | Type |
|---------|--------|-----------|------|
| SST-2 | DeepSeek-ICL | 1,820 | Recurring (Per Inference) |
| | **StructCBM (Ours)** | **75** | **One-time (Construction)** |
| AGNews | DeepSeek-ICL | 7,600 | Recurring (Per Inference) |
| | **StructCBM (Ours)** | **173** | **One-time (Construction)** |

### A.8.3 SCALABILITY STRATEGY FOR LARGE-SCALE SCENARIOS ($N \gg 10$)

Although the unit costs are stable, the total number of pairwise comparisons for discriminative concepts ($\mathcal{C}_d$) grows quadratically ($O(N^2)$) with the number of classes $N$. To prevent prohibitive API demands in scenarios with hundreds of classes, we propose a **Similarity-based Pre-filtering Strategy**.

**Mechanism.** Instead of exhaustively generating $\mathcal{C}_d$ for all $N(N-1)$ pairs, we utilize the representations of the Prototypical Concepts ($\mathcal{C}_p$)—which are generated linearly ($O(N)$)—to map the semantic topology of the labels.

1) **Cluster Candidates:** We calculate the cosine similarity between the embeddings of $\mathcal{C}_p$ for all classes.

2) **Filter Pairs:** For each target class $i$, we identify the top-$K$ most semantically similar classes (distractors).

3) **Targeted Generation:** We generate discriminative concepts $\mathcal{C}_d$ *only* for these specific top-$K$ pairs.

**Complexity Analysis.** This strategy effectively prunes the search space. Because the filtering process relies on local embedding similarity (zero API cost), the major API cost is restricted to the targeted generation. For $N$ classes with a neighbor size of $K$, the complexity is reduced from $O(N^2)$ to $O(NK)$. Since $K$ (confusing classes) is typically small and independent of $N$, this ensures linear scalability.

