# OpenReview forum: "Adaptive Concept Discovery for Interpretable Few-Shot Text Classification"
_ICLR.cc/2026/Conference — ICLR 2026 Poster_

### Official Review · Reviewer_mVk8 · 2025-10-16

**Soundness:** 2
**Presentation:** 3
**Contribution:** 2
**Rating:** 4
**Confidence:** 3

**Summary:**

This paper proposes an interpretable few-shot text classification framework based on a Concept Bottleneck Model (CBM) paradigm. The approach introduces a separation of concepts into prototypical and discriminative sets, and leverages Large Language Models (LLMs) to dynamically generate concept proposals for both. Furthermore, a two-stage prediction process is proposed, consisting of prototype pruning followed by re-ranking of discriminative concepts. The overall framework aims to improve interpretability and adaptability in few-shot scenarios.

**Strengths:**

1. Clarity and readability: The paper is well written and easy to follow, with a logical presentation of motivation, methodology, and results.

2. Novel paradigm: The use of LLMs to dynamically generate and structure concepts into two interpretable sets (prototypical vs. discriminative) is interesting and aligns with the goal of interpretable few-shot learning.

3. Interpretable design: The combination of concept separation and the two-stage inference mechanism provides a clear and interpretable reasoning process.

**Weaknesses:**

1. Over-reliance on LLM-generated concepts:
The proposed approach heavily depends on the reliability of concepts generated by LLMs. Given the limited few-shot data, the LLM-generated concepts and the closed-loop feedback mechanism may introduce noise or bias. The poor performance on SST-2 (Table 3) supports this concern.

2. Limited evaluation scope:
The experimental comparison is restricted to only two white-box baselines. It would strengthen the paper to include more interpretable or concept-based baselines, especially those already listed in Table 1.

3. Lack of discussion on few-shot text classification challenges:
Few-shot text classification is inherently ill-posed, and LLMs are known to perform well in such settings even without explicit concept modeling. The experimental results from black-box methods reinforce this observation, but the paper lacks a sufficient discussion or justification on why the CBM approach is preferable in this context.

4. Insufficient evaluation diversity:
Experiments are conducted only on the 10-shot setting. Including 1-shot and 5-shot results would provide a more comprehensive understanding of the model’s robustness under varying data constraints.

5. Unclear concept definitions:
The definitions of prototypical and discriminative concept sets remain somewhat abstract. Clearer explanations and concrete examples (e.g., illustrating how a sentence or phrase maps to each set) would greatly improve the reader’s understanding of how the model differentiates between the two concept types.

**Questions:**

1. Can the authors provide examples to illustrate the distinction between prototypical and discriminative concept sets?

2. How does the model handle unreliable or irrelevant concepts generated by the LLM, especially under few-shot conditions?

3. Could the proposed method generalize to fewer samples (e.g., 1-shot or 5-shot)? If not, what modifications would be necessary?

4. Minor: How many classes are there for each task used in the experiments?

---

> ### Author Response · Authors · 2025-11-23
>
> We thank the reviewer for the effort in reviewing our paper. We appreciate the recognition of our **clear presentation, novel paradigm, and interpretable method design**. Our replies to the concerns are presented below. We will prepare a revised version of the paper accordingly.
>
> ---
>
> **mVk8-W1,Q2**: The proposed method relies heavily on the quality of the constructed concepts. How can the proposed method avoid introducing noise or bias from low-quality concepts, especially on the SST2 dataset? How does the model handle unreliable or irrelevant concepts generated by the LLM, especially under few-shot conditions?
>
> **Reply to mVk8-W1,Q2**:
>
> We thank the reviewer for this fundamental question regarding the reliability of LLM-generated concepts. We understand the concern that relying on LLMs might introduce noise or bias. However, we contend that utilizing LLMs for concept generation is an established paradigm in recent CBM research (e.g., TBM [1], $C^3M$ [2]) and is a strategic necessity in few-shot settings. In a few-shot scenario, the primary bottleneck is the lack of a supervision signal rather than the presence of noise. By utilizing LLM-generated concepts, we introduce a strong, "world knowledge" prior that effectively brings down the supervision requirements.
>
> To support our claim on the lack of data, we add black-box models that are finetuned on few-shot data as baselines. We select RoBERTa and DistilBERT as baselines, since they have a similar level of parameters as our method. Results show that these models perform worse than our method, confirming that the main challenge is data scarcity. Detailed results are shown in the **Reply to mVk8-W3**.
>
> Furthermore, our framework does not blindly accept LLM outputs; it actively mitigates unreliable concepts through a two-stage process:
> 1. **Filtration during the Annotation**: The LLM annotation step at the beginning of concept construction acts as a gatekeeper to **discard irrelevant concepts** before they enter the system.
> 2. **Alignment during the Tuning**: Our "Concept Tuning" loop resolves representation mismatches. This aligns the vector representation of the concept to the actual data, and guarantees that the **minor noises are removed** from the textual concepts to fit the ground truth data.
>
> Regarding the specific performance drop on the SST-2 dataset, our analysis identified the cause **not as low-quality concepts**, but as **"over-correction"** during the tuning phase due to loose constraints. In binary classification, discriminative concepts often implicitly support the positive class. Our original tuning aggressively pushed concepts away from error samples but lacked sufficient force to keep them anchored to positive samples. To resolve this, we implemented a stricter regularization constraint. During the refinement of Discriminative Concepts ($C_d$), we now enforce that the concept's average similarity to the positive samples must not decrease while it is being adjusted to fix an error.
>
> As shown in the table below, this improved regularization successfully stabilized the tuning process and resulted in performance gains on SST-2. This confirms that the issue was a controllable optimization problem rather than a fundamental flaw in the quality of the concepts.
>
> *Effectiveness of Regularization*
> | Dataset | Refinement Method | Accuracy (%) |
> |---------|---|---|
> | SST2  | No Refine | 80.29%         |
> | SST2  | Refine w/o Reg. |  76.77%    |
> | SST2  | Refine w/ Reg. | 81.27%      |
>
> ---
>
> **mVk8-W2**: More white-box baselines are suggested to be added.
>
> **Reply to mVk8-W2**:
> We sincerely appreciate the reviewer for the suggestion. We have included additional white-box baselines, C3M [2] and SparseCBM [3] , in our experiments. The results are presented in the following table. Our method outperforms these baselines, demonstrating its effectiveness.
>
> *Comparison with additional white-box baselines*
> | Method | MedAbs | AGNews |
> |---------|--------------|---|
> | DeepSeek-ICL    | 63.74 | 79.00 |
> | CB-LLM    | 35.21 | 68.34 |
> | C3M    | 31.09 | 55.68 |
> | SparseCBM    | 37.78 | 62.84 |
> | StructCBM(Ours)    | 60.11 | 81.58 |

---

> ### Author Response · Authors · 2025-11-23
>
> **mVk8-W3**: Lack of justification on few-shot text classification challenges.
>
> **Reply to mVk8-W3**:
> We respectfully suggest that there may be a misunderstanding regarding our problem setting. Our work focuses on a highly practical and challenging scenario: how to perform interpretable classification on massive amounts of unlabeled text using only a handful of labeled examples. This constraint is critical in high-stakes domains such as Finance, E-commerce, and Medical diagnosis, where data privacy, labeling costs, and the need for transparency are paramount.
>
> In this specific scenario, black-box models face critical limitations despite their general capabilities. While Large Language Models (LLMs) demonstrate strong few-shot performance, their **inference latency and computational costs** are prohibitive when applied to massive-scale testing corpora, making them economically and computationally infeasible for high-throughput real-world applications [1,2,3]. Conversely, directly fine-tuning smaller black-box models (e.g., RoBERTa) on few-shot data reduces inference costs, but as shown in our experimental results, their performance degrades under data scarcity, failing to meet the accuracy requirements of the task.
>
> Concept Bottleneck Models (CBMs) offer a unique solution by balancing high interpretability with lightweight inference, potentially achieving high performance without the overhead of LLMs. However, existing CBM approaches have largely overlooked the constraints of the few-shot setting and are limited in effectiveness.
>
> Therefore, since neither direct LLM inference nor standard small-model fine-tuning provides a viable solution for the "Massive Unlabeled + Few-Shot Labeled + Interpretable" constraint, we propose StructCBM to bridge this gap, systematically solving the challenges of adapting interpretable concept bottlenecks to few-shot environments.
>
>
> *Comparison with small black-box baselines*
> | Method | SST2 | MedAbs |
> |---|---|---|
> | RoBERTa    | 50.08 | 51.70 |
> | DistilBert    | 58.59 | 43.49 |
> | StructCBM(Ours)    | 80.29 | 60.11 |
>
> ---
>
> **mVk8-W4, Q3** Including 1-shot and 5-shot results would provide a more comprehensive understanding of the model’s robustness under varying data constraints.
>
> **Reply to mVk8-W4, Q3**: We appreciate the reviewer for the suggestion. The additional comparison are shown in the following table, our method consistently outperforms baselines across all settings (1, 5, and 10 shots). It can be seen that with the increase of avaliable samples, our method becomes increasingly better. And the performance difference between 5 and 10 shot setting are relatively small, which indicates the robustness of our method in few-shot scenarios.
>
> *Experiments on different few-shot settings*
> | Dataset | Shot-number | Accuracy (%) |
> |---------|------------ | --- |
> |AGNews   | 1          | 65.71 |
> |AGNews   | 5        |  80.76 |
> |AGNews| 10        | 81.58 |
> |FinaQuery| 1          |   58.73 |
> |FinaQuery| 5        |   73.60 |
> |FinaQuery| 10        | 77.42 |

---

> ### Author Response · Authors · 2025-11-23
>
> **mVk8-W5, Q1** Clearer explanations of the definitions of prototypical and discriminative concept sets are needed. Concrete examples would be helpful.
>
>
> **Reply to mVk8-W5, Q1**: We would love to clarify our definitions on the prototypical and discriminative concept sets. Related information can also be found in Section 3.1 and illustrated in Figure 4 of our paper.
>
> Our framework distinguishes between two sets of concepts based on their specific roles in the decision-making process. Prototypical Concepts ($C_p$) serve as the foundation of the model's understanding. They capture universal commonalities present in the majority of samples for a specific target class, ensuring that the model correctly identifies the input's core topic.
>
> In contrast, Discriminative Concepts ($C_d$) act as exclusionary filters. While they capture features relevant to the target class, their primary function is to distinguish that target from a competing label, referred to as a "Distractor." These concepts refine the classification process by explicitly ruling out categories that are semantically similar but ultimately incorrect.
>
> To ensure the model captures rich and nuanced semantics, every concept within the framework must adhere to a specific structure consisting of a Name and a Detailed Description. The Name provides a concise title, while the Detailed Description offers a comprehensive explanation of the concept's boundaries and specific meaning. In implementation, both components are encoded into embeddings, which are then utilized to calculate semantic similarity with the input text.
>
> Here, we present concept examples from FinaQuery dataset to better illustrate our definitions (Target Label: Budgeting & Cash Flow Management):
>
> - Prototypical Concept Example:
>     - Concept: "Income-expense ratio analysis"
>     - Description: "Systematic evaluation of the proportional relationship between earned income and recurring expenses, particularly housing costs, to assess financial sustainability and affordability thresholds"
>     - Description: "The process of evaluating the proportion of operational revenue against recurring expenditures to determine financial liquidity."Role: Ensures the model identifies the core topic.
>
> - Discriminative Concept Example:
>     - Scenario A (vs. Distractor: Debt Management): The model generates concepts to explicitly exclude credit-related contexts.
>       - Concept: "Future expense planning without borrowing."
>       - Description:"Discussions about anticipating and preparing for upcoming expenses, major life changes, or financial goals without seeking or mentioning credit options, loans, or borrowing strategies. Includes planning for housing changes, education costs, vehicle maintenance, or lifestyle adjustments using existing resources"
>     - Scenario B (vs. Distractor: Savings & Emergency Funds):The model focuses on active flow management rather than accumulation.
>       - Concept: "Recurring expense management."
>       - Description: "Posts about managing, negotiating, or canceling regular monthly payments, subscription services, or ongoing financial commitments, often with concerns about contractual obligations or cancellation difficulties"
>
> ---
>
> **mVk8-Q4**: How many classes are there for each task used in the experiments?
>
> **Reply to mVk8-Q4**: The class numbers of all datasets are shown in the table below.
>
> | Dataset | SST2 | MedAbs | AGNews | FinaQuery |
> |---|---|---|---|---|
> | Class Number | 2 | 5 | 4 | 8 |
>
> ---
>
> - [1] Interpretable-by-Design Text Understanding with Iteratively Generated Concept Bottleneck
> - [2] Interpreting Pretrained Language Models via Concept Bottlenecks
> - [3] Sparsity-Guided Holistic Explanation for LLMs with Interpretable Inference-Time Intervention

---

### Official Review · Reviewer_GAko · 2025-10-30

**Soundness:** 3
**Presentation:** 4
**Contribution:** 3
**Rating:** 4
**Confidence:** 3

**Summary:**

StructCBM is a novel Concept Bottleneck Model (CBM) framework designed to overcome the high inference costs and "black-box" nature of Large Language Models (LLMs) in few-shot text classification. StructCBM introduces a new CBM paradigm that relies solely on sample-concept similarity for prediction, enabling a lightweight, LLM-free inference process. Experiments show that StructCBM surpasses prior CBMs and achieves performance comparable to LLMs.

**Strengths:**

The StructCBM framework introduces a novel paradigm for Concept Bottleneck Models (CBMs) that addresses the primary limitations of Large Language Models (LLMs)—high inference costs and lack of interpretability—while overcoming the challenges faced by prior CBM approaches in few-shot learning scenarios. The method achieves performance comparable to powerful black-box LLMs. Notably, on semantically rich datasets like AGNews, StructCBM surpasses the 10-shot Deepseek-ICL baseline, and on MedAbs, its performance is comparable to the zero-shot Deepseek-Direct.

**Weaknesses:**

1. The paper clearly emphasizes that StructCBM's strength lies in its LLM-free, lightweight inference process. However, the pairwise formulation of Discriminative Concepts (Cd) scales quadratically with the number of classes, which is noted as a significant challenge for tasks with large label sets.

2.The final step involves fine-tuning the embedding model e(⋅) to improve matching performance. Crucially, the fine-tuning data P (Equation 5) is constructed only using the prototypical concepts (Cp) and associated samples, explicitly excluding Cd.  More ablation study is suggested to compare performances when Cd was included in the fine-tuning set.

**Questions:**

1. Regarding the quadratic scaling of Cd, the paper suggests exploring "on-demand concept generation" for only the most confusable class pairs as a solution. How would StructCBM implement this on-demand concept selection or generation in a practical, real-world inference scenario? Does identifying the "most confusable" pairs inherently require a costly preliminary analysis that might undermine the overall efficiency gain?

2.The Concept Tuning stage (generate-predict-refine workflow)—is critical for discovering higher-quality concepts. Yet, the ablation study showed that this refinement process negatively impacted performance on the semantically sparse SST2 dataset, which was hypothesized to be due to overfitting. So what practical criteria or internal heuristics were considered or could be developed to automatically detect when concept refinement, particularly the Semantic Tuning process (Algorithm 1), is beginning to induce overfitting on semantically poor data (like SST2)? Could a metric reflecting concept redundancy or semantic density be incorporated into the refinement trigger to ensure robustness across diverse datasets.

---

> ### Author Response · Authors · 2025-11-23
>
> We thank the reviewer for the effort in reviewing our paper. We appreciate the recognition of our **novel paradigm and comparable performance to powerful black-box LLMs**. Our replies to the concerns are presented below. We will prepare a revised version of the paper accordingly.
>
> ---
>
> **GAko-W1, Q1**: The formulation of $C_d$ may have difficulty in scaling to large class scenarios. How would the mentioned 'on-demand concept selection' method address such a problem? And what is the related cost?
>
> **Reply to GAko-W1, Q1**: We appreciate the reviewer for highlighting the scalability of $C_d$ in large-scale scenarios. This aligns with the scalability discussion we briefly noted in our limitations section. While the full implementation of these large-scale strategies extends beyond the primary scope of this work, we are happy to share the concrete logic and cost analysis of the "on-demand" mechanism we have designed.
>
> To resolve the quadratic cost of exhaustive generation ($O(N^2)$ for $N$ classes), we propose the **similarity-based pre-filtering** strategy:
>
> - Mechanism: After generating $C_p$ (whose complexity is $O(N)$), we use the representations of $C_p$ to evaluate the similarity between classes. Then, for each class, its $C_d$ is only generated for its top-$K$ similar classes. This avoids unnecessary comparisons between semantically distant topics (e.g., Sports vs. Finance) and improves efficiency.
>
> - Related Cost: This strategy reduces the search space from global $O(N^2)$ to $O(NK)$. Because the filtering process does not require LLM calls, the major cost lies in the $C_d$ generation. For $N$ classes, each class should compare with $K$ candidates, which results in the $O(NK)$ complexity.
>
> For a more comprehensive analysis of our computational cost, please refer to our **Reply to XLoV-W1, W2, Q1, Q2**.
>
> ---
>
> **GAko-W2**: When fine-tuning the embedding model, how is the performance of adding $C_d$ into the construction of fine-tuning dataset?
>
> **Reply to GAko-W2**: We sincerely appreciate the reviewer for bringing up this suggestion. We have conducted supplementary experiments to examine the performance of how adding $C_d$ into the construction of the finetuning dataset affects the final performance. The results are presented in the following table. Results show that adding $C_d$ can further improve the performance, yet the improvements are relatively subtle. This is because $C_p$ focuses on general and prototypical knowledge, which is more effective in improving the training performance.
>
> *Ablation study on finetuning dataset construction*
>
> | Dataset | Finetuning Data | Accuracy (%) |
> |---|---|---|
> |AGNews| None         |  81.41  |
> |AGNews| Only Cp         | 81.58 |
> |AGNews| Cp + Cd       | 82.05  |
> |MedAbs| None         | 59.00 |
> |MedAbs| Only Cp         | 60.11 |
> |MedAbs| Cp + Cd       | 60.70 |

---

> ### Author Response · Authors · 2025-11-23
>
> **GAko-Q2**: The refinement process may lead to overfitting problems, what are possible solutions to address such problem? Especially, will the Semantic Tuning phase induce overfitting on semantically poor data like SST2? Could a metric reflecting concept redundancy or semantic density be incorporated into the refinement trigger to ensure robustness across diverse datasets?
>
> **Reply to GAko-Q2**: We sincerely appreciate the reviewer for inspiring analysis. Our replies are detailed as follows.
>
> **Solution to the Overfitting Problem**
>
> For the overfitting problem that appeared at SST2, we have incorporated regularization objectives into our current framework. Specifically, during the refinement phase for Discriminative Concepts ($C_d$), we introduced a stricter constraint: in addition to distancing the concept from the negative samples, we now enforce that the concept's average similarity to the positive samples must not decrease. As hypothesized and demonstrated in the table, by enforcing the average-similarity constraint (w/ Reg.), the model achieves stability and reaches a peak accuracy of 81.27% on SST2 dataset, surpassing the results with no refinement.
>
> *Effectiveness of Regularization*
>
> | Dataset | Refinement Method | Accuracy (%) |
> |---------|---|---|
> | SST2  | No Refine | 80.29         |
> | SST2  | Refine w/o Reg. |  76.77    |
> | SST2  | Refine w/ Reg. | 81.27      |
>
> **Discussion on Semantic Tuning**
>
> Based on the regularization constraint introduced in the last paragraph, the overfitting risk introduced by semantic tuning can be well mitigated. In fact, the semantic limitation of SST2 that caused overfitting mainly comes from its binary classification nature. The ‘positive’ and ‘negative’ labels must be distinctly separated. However, our previous implementation loosened such a requirement, resulting in adding concepts with vague polarity and caused overfitting. Now such an issue is solved.
>
> **Metric to Monitor Overfitting Issue**
>
> To mitigate the concept redundancy during tuning, the concept polarity to samples can be a good metric. Essentially, when a concept is similar to a certain sample but dissimilar to the majority of all samples from the same class, it could lead to overfitting. A formal definition of concept polarity is as follows.
>
> For a set of samples $X$ from the same class, when tuning a concept $c$ for a sample $x \in X$, we note the similarity between $c$ and $x$ as $s_{spl}(c,x) = sim(c,x)$ and the average similarity between $c$ and $X$ as
> $s_{set}(c,X) = \frac{1}{|X|-1} \sum_{x' \in X; x' \neq x} sim(c,x')$.
> Then the concept polarity $p(c)$ is calculated as:
>
> $$p(c)=\frac{s_{spl}(c,x)}{s_{set}(c,X)}$$.
>
> When $p(c)$ is larger than a threshold (e.g., $p(c)>1.2$), it reflects a stronger polarity to a certain sample than the general samples, and thus can be pruned.

---

### Official Review · Reviewer_swGd · 2025-10-31

**Soundness:** 3
**Presentation:** 3
**Contribution:** 3
**Rating:** 6
**Confidence:** 4

**Summary:**

This paper presents StructCBM, an LLM-augmented Concept Bottleneck Model (CBM) for interpretable few-shot text classification. Unlike traditional CBMs that require large datasets and trained classifiers, StructCBM predicts labels through sample–concept similarity, removing the need for parametric training. It introduces a dual-level concept structure: prototypical concepts (Cp) capture representative class features for recall, while discriminative concepts (Cd) distinguish confusable classes for precision. A generate–predict–refine loop further improves concept quality through LLM-based semantic and logical refinements, while inference relies purely on embedding similarity without invoking an LLM. Experiments on SST-2, AGNews, MedAbs, and FinaQuery under a 10-shot setting show that StructCBM consistently outperforms prior CBMs such as TBM and CB-LLM, achieving performance close to black-box LLM baselines like DeepSeek-ICL, with the additional benefits of interpretability and computational efficiency.

**Strengths:**

- S1: The paper explores an underexamined intersection of interpretability and few-shot learning, where existing LLM-based CBMs have been ineffective.

- S2: The two-stage architecture (prototypical vs. discriminative concepts) introduces a clear and interpretable reasoning process, akin to human recall–then–refine reasoning. The closed-loop refinement mechanism adds adaptivity and resilience in low-data regimes. Its non-parametric design avoids fine-tuning while maintaining transparency and efficiency.

- S3: Experiments across diverse domains show consistent and significant gains over interpretable baselines. The ablation and sensitivity studies clearly isolate each component’s contribution. The qualitative case study effectively demonstrates interpretability and transparent reasoning.

- S4: Inference is entirely LLM-free, yielding substantial computational savings compared to other LLM-augmented CBMs. Each prediction is explicitly grounded in human-readable concepts, supporting transparency and trustworthiness.

**Weaknesses:**

- W1: The pairwise Cd formulation scales quadratically with the number of classes, limiting applicability to large-label tasks. While acknowledged by the authors, there is no quantitative analysis of computational or memory overhead.

- W2: Although inference is lightweight, concept generation and refinement heavily depend on LLM capabilities. The paper does not assess the impact of different LLMs (e.g., GPT-4 vs. DeepSeek-V3), leaving robustness across models unclear.

- W3: The evaluation is restricted to 10-shot experiments, without studying performance trends as data increases. Some baselines (e.g., TBM, CB-LLM) may be under-optimized for few-shot scenarios, possibly skewing comparisons. Interpretability is validated only qualitatively, with no quantitative concept fidelity or human evaluation.

- W4: The generate–predict–refine loop lacks analysis of computational cost, convergence, and stopping criteria, leaving uncertainty about deployment feasibility.

- W5: The paper claims that code is released at an anonymous link, but the provided URL returns “The requested file is not found.”

**Questions:**

n/a

---

> ### Author Response · Authors · 2025-11-23
>
> We thank the reviewer for the effort in reviewing our paper. We appreciate the recognition of our **novel problem, clear and reasonable method design, diverse and effective experiments, and high efficiency and interpretability**. Our replies to the concerns are presented below. We will prepare a revised version of the paper accordingly.
>
> ---
>
> **swGd-W1**: Additional quantitative analysis of the computational or memory overhead brought by the pairwise Cd formulation is suggested, especially in numerous class scenarios.
>
> **Reply to swGd-W1**: We appreciate the reviewer’s suggestion to quantify the overhead of the pairwise formulation. We have analyzed the computational and memory costs, specifically focusing on the $C_d$ component.
>
> 1. **Quantitative Analysis of $C_d$ Computation**: While the pairwise formulation theoretically scales quadratically ($O(N^2)$ for $N$ classes) in an exhaustive setting, our empirical data shows that the **unit cost** per class remains manageable for standard benchmarks due to efficient stopping criteria. Based on our experiments across datasets (SST-2, AGNews, FinancialQueries), we quantified the average API calls required specifically for $C_d$ generation in the table below. On average, generating discriminative concepts for a class (against its distractors) requires only ~10 API calls in a 10-shot setting. This confirms that for moderate-sized datasets, the cost is highly controllable.
>
> |  Per class | 1 shot |  5 shot | 10 shot |
> |---|---| --- | --- |
> | Cd    | 4.99   |  9.38  |  9.08 |
>
> 2. **Scalability in "Numerous Class" Scenarios**: We acknowledge that for scenarios with hundreds of classes, the exhaustive $O(N^2)$ generation for $C_d$ becomes the bottleneck. We thus propose the **similarity-based pre-filtering** strategy, whose details are as follows:
>
>    - Mechanism: After generating $C_p$, we use the representations of $C_p$ to evaluate the similarity between classes. Then for each class, its $C_d$ is only generated for its top-$K$ similar classes. This avoids unnecessary comparisons between semantically distant topics (e.g., Sports vs. Finance) and improves efficiency.
>
>    - Related Cost: This strategy reduces the search space from global $O(N^2)$ to $O(NK)$. Because the filtering process does not require LLM calls, the major cost lies in the $C_d$ generation. For $N$ classes, each class should compare with $K$ candidates, which results in the $O(NK)$ complexity.
>
> 3. **Memory Overhead**: Regarding memory, the overhead brought by additional $C_d$ concepts is negligible.
>    - Storage: Even with hundreds of classes, storing the corresponding concept embeddings requires <10MB, yet a language embedding model backbone generally requires ~$10^2$ MB.
>    - Inference: The memory footprint during inference is constant and independent of the total number of $C_d$ pairs, as we only retrieve the fixed Top-$K$ concepts for the input context.
>
> ---
>
> **swGd-W2**: The ability of LLM is essential to the effectiveness of the method, additional analysis on how different LLMs affect the final performance is suggested.
>
> **Reply to swGd-W2**: We include an ablation study on different choices of LLMs for concept annotation. Considering the ability of the exsiting LLM ``DeepSeek-V3``, we select ``GPT-4o`` and ``Qwen3-Plus`` for higher-level and lower-level comparisons. The results are listed in the table below. The results indicate that a better LLM generally leads to improved performance, and our method is generally robust to different LLM choices.
>
> *Evaluation on Different LLMs*
> | Dataset | LLM | Accuracy |
> |---------|--------------|---|
> | SST2    | DeepSeek |80.29|
> | SST2    | GPT-4o |86.38 |
> | SST2    | Qwen3-Plus | 79.63 |
> | AGNews    | DeepSeek |81.58|
> | AGNews    | GPT-4o |85.07|
> | AGNews    | Qwen3-Plus |81.76|

---

> ### Author Response · Authors · 2025-11-23
>
> **swGd-W3**: 1): Additional analysis on different shots of data is suggested to better validate the robustness of the method in few-shot scenarios. 2): Some baselines (TBM, CB-LLM) are under-optimized for few-shot scenarios. 3): Quantitative interpretability analysis on the concept fidelity or human evaluation is suggested.
>
> **Reply to swGd-W3**:
>
> 1. Following the suggestion from the reviewer, we add experiments on 1-shot and 5-shot settings to validate the robustness of our method in few-shot scenarios. The results are presented in the table below. It can be seen that with the increase in available samples, our method becomes increasingly better. And the performance difference between 5 and 10 shot settings is relatively small, which indicates the robustness of our method in few-shot scenarios.
>
> *Experiments on different few-shot settings*
> | Dataset | Shot-number | Accuracy (%) |
> |---------|------------ | --- |
> |AGNews   | 1          | 65.71 |
> |AGNews   | 5        |  80.76 |
> |AGNews| 10        | 81.58 |
> |FinaQuery| 1          |   58.73 |
> |FinaQuery| 5        |   73.60 |
> |FinaQuery| 10        | 77.42 |
>
> 2. The comparison with TBM and CB-LLM is not skewed. Their ineffectiveness in handling a few-shot scenario is precisely the motivation for our work, and the inferior results validate our hypothesis. To ensure a comprehensive and fair comparison, we also added LLM performances to show the upper-bound performance on the few-shot tasks. And our closeness to LLM performance further validates the effectiveness of our method.
>
> 3. We sincerely appreciate the reviewer for mentioning this problem. To ensure that the LLM's internal logic aligns with human reasoning, we initiated a human evaluation study. We will recruit volunteers to examine 1) whether the annotation from LLM aligns with human and 2) whether the concepts and their related texts match human understanding. This evaluation is currently in process, and the results will be reported within the rebuttal period.

---

> ### Author Response · Authors · 2025-11-23
>
> **swGd-W4**: The generate–predict–refine loop lacks analysis of computational cost, convergence, and stopping criteria, leaving uncertainty about deployment feasibility.
>
> **Reply to swGd-W4**: We sincerely appreciate the reviewer for pointing out the need for a deeper analysis of the loop's behavior. We agree that analyzing the stopping criteria and convergence is critical for assessing deployment feasibility. Below, we conduct a detailed analysis to quantify the cost and stability of the "Generate-Predict-Refine" loop. We clarify that this loop is a controlled offline process with strict bounds, ensuring deployment feasibility.
>
> 1. **Computational Cost Analysis**: To address the uncertainty about cost, we quantified the API consumption required to complete this loop. Based on our experiments (across SST-2, AGNews, and FinaQuery), the Average Unit Cost to run the full loop per class is shown below. (Remark: The cost per class remains stable across datasets, confirming the predictability of our method.)
>
> *Average API calls per class on the dataset*
> |  Per class | 1 shot |  5 shot | 10 shot |
> |---------|--------------| --- | --- |
> | Cp    |         4.83        | 11.12 | 15.38 |
> | Cd    |             4.07   |  9.38  |  9.08 |
> | Tuning    |           0         | 9.77 | 13.01 |
>
> 2. **Explicit Stopping Criteria**: Our framework does not rely on open-ended generation. We implement deterministic criteria for each stage to guarantee termination:
>    - Stage 1: Concept Generation
>      - For $C_p$ (Coverage-based): The loop terminates when every sample in the support set is covered by at least one valid concept (verified by LLM annotation).
>      - For $C_d$ (Validity-based): The loop terminates when the generated concept $c_{i,j}$ satisfies the Discriminative Condition (as detailed in Section 3.1): $Score(\text{target}) > 0$ AND $Score(\text{distractor}) \le 0$.
>      - Implementation Detail: Empirically, thanks to the strong instruction-following capabilities of LLMs, this process converges rapidly. In extremely low-shot settings (e.g., 1-shot), we enforce a minimum requirement of 3 concepts to ensure robustness, though this threshold becomes less critical as the number of shots increases (e.g., 10-shot).
>    - Stage 2: Concept Tuning
>      - The refinement loop stops when either: (1) The classification accuracy on the support set stabilizes (improvement $< \epsilon$), meaning all fixable errors are resolved; or (2) A pre-set maximum number of tuning rounds (e.g., 5 rounds) is reached.
>      - Regularization Impact: Without regularization, the loop converges almost immediately as the LLM generates concepts that perfectly fit the samples. When strict regularization is applied, the rejection rate for new concepts may increase (leading to slightly higher API usage during semantic refining). However, since we enforce a maximum round limit, the overall tuning phase remains strictly controlled.
> 3. **Convergence Analysis**: We empirically validated the convergence of this loop following the criteria above. As shown in the Cost Analysis Table, the average number of API calls for tuning is relatively low (~6.24 calls per class in 10-shot), proving efficient convergence.
>    - Why it converges:
>      - Generation Phase: Due to the limited size of the few-shot support set and the strong semantic reasoning of modern LLMs, the model identifies covering concepts within very few iterations.
>      - Tuning Phase: The "extraction-based" refinement acts as a discrete optimization step. While strict regularization might trigger retries, the process is structurally bounded. The low average call count indicates that the model typically snaps to an optimal representation quickly (within 2-3 rounds) rather than oscillating indefinitely.
> 4. **Deployment Feasibility**: Finally, we address the concern regarding real-time usage:
>    - Offline vs. Online: This entire loop is a one-time offline construction process.
>    - Deployment: Once the loop satisfies the stopping criteria, the concept library is frozen. The deployed model functions as a static retriever and classifier, incurring zero additional computational cost or latency compared to standard inference.
>
> ---
>
> **swGd-W5**: The link to the code is currently inaccessible.
>
> **Reply to swGd-W5**: We have double-checked the code link; the reviewer can access it through the link on the ICLR website, which is not modified since submission. And the link in the pdf will be corrected in a revised version.

---

### Official Review · Reviewer_XLoV · 2025-10-31

**Soundness:** 3
**Presentation:** 3
**Contribution:** 3
**Rating:** 6
**Confidence:** 3

**Summary:**

This paper proposes StructCBM, a novel framework for interpretable few-shot text classification that leverages an adaptive concept library. The core idea is to decompose the classification task into a two-stage process: (1) generating and maintaining a library of human-readable prototypical (positive) and discriminative (negative) concepts for each class using a Large Language Model (LLM), and (2) making predictions by measuring the similarity between an input sample and these concepts. The framework's key innovation lies in its closed-loop "Concept Tuning" mechanism. This mechanism uses misclassified samples as feedback to iteratively refine the concept library through Semantic Tuning (rewriting existing concepts to fix recall or ranking errors) and Logical Tuning (generating entirely new concepts for hard samples). Finally, the framework fine-tunes an embedding model to better align sample and concept representations. The authors evaluate StructCBM on four diverse text classification benchmarks (SST2, AGNews, MedAbs, FinaQuery) in a 10-shot setting, demonstrating superior performance over other white-box and concept-based baselines while maintaining full interpretability.

**Strengths:**

-**Novel Adaptive Mechanism**: The Concept Tuning stage is the paper's standout feature. By using classification errors as direct feedback to improve the concept library, the framework creates a powerful closed-loop system that can self-correct and evolve, which is a significant step forward from static concept bottleneck models.

-**Strong Empirical Validation**: The experiments are comprehensive, covering four distinct domains. The results clearly show that StructCBM outperforms other white-box and concept-based methods by a large margin, while remaining competitive with powerful black-box LLM approaches (Deepseek). This convincingly demonstrates the framework's efficacy.

-**Preserved Interpretability**: The paper successfully achieves its primary goal of interpretability. Every prediction is grounded in explicit, human-readable concepts, and the tuning process itself is designed to improve the semantic quality of these concepts.

**Weaknesses:**

-**Scalability to Many Classes**: The paper focuses on benchmarks with a relatively small number of classes (e.g., SST2 has 2, AGNews has 4). It is unclear how the concept library and the tuning process would scale to problems with dozens or hundreds of classes. The cost of maintaining and tuning a large number of prototypical and discriminative concepts per class could become prohibitive.

-**LLM Dependency and Cost**: The framework relies heavily on an LLM for concept generation and rewriting. The paper does not discuss the computational cost or latency implications of these LLM calls, which is a critical factor for real-world deployment. It would be useful to know the number of LLM calls required per tuning cycle.

-**Lack of Human Evaluation**: While the concepts are human-readable, the paper provides no human evaluation to assess their actual quality, coherence, or usefulness to an end-user. An expert or crowd-sourced study on the concepts' validity would strengthen the interpretability claim.

-**Black-box Baseline Choice**: The black-box baselines (Deepseek-Direct/ICL) are strong, but it would be more compelling to also compare against a fine-tuned smaller model (e.g., a fine-tuned DistilBERT on the 10-shot data) to better isolate the benefit of the concept-based approach versus just using a more powerful LLM.

**Questions:**

-**Scalability**: How does the runtime and memory footprint of StructCBM scale with the number of classes? Have you experimented with datasets that have a larger number of categories (e.g., 10+)? What are the bottlenecks?

-**LLM Cost**: Can you provide an estimate of the number of LLM calls (for both initial generation and tuning) required per dataset? How sensitive is the final performance to the quality/size of the LLM used for concept engineering?

-**Human Evaluation**: Have you conducted any user studies to validate that the generated and tuned concepts are indeed meaningful, non-redundant, and helpful for a human to understand the model's decisions? This seems crucial for a paper centered on interpretability.

-**Ablation on Tuning**: The paper mentions evaluating the contribution of each component. Could you share the performance drop when the Semantic Tuning and Logical Tuning stages are ablated? This would help quantify the value of the core adaptive mechanism.

---

> ### Author Response · Authors · 2025-11-23
>
> We thank the reviewer for the effort in reviewing our paper. We appreciate the recognition of our **novel adaptive mechanism, strong empirical validation, and preserved interpretability**. Our replies to the concerns are presented below. We will prepare a revised version of the paper accordingly.
>
> ---
>
> **XLoV-W1, W2, Q1, Q2**: More discussion on the scalability, LLM cost, and efficiency perspectives of the method on numerous classes.
>
> **Reply to XLoV-W1, W2, Q1, Q2**:
> We address the concerns regarding scalability, LLM costs, and efficiency together, as they are intrinsically linked.
>
> 1. **Summary**: The primary cost of our framework stems from API calls during the offline construction phase. Our analysis shows that this cost is structurally predictable: the API consumption per class and per pair of classes remains stable across datasets. While total costs increase with class count, the unit cost does not explode. Furthermore, for large-scale scenarios ($N \gg 10$), we propose concrete pruning strategies to mitigate the bottleneck.
>
> 2. **Unit Cost Analysis**: From Few to Many Classes, we assess scalability via a "Unit Cost" perspective. Since API usage is driven by strict quality-control mechanisms (stopping criteria), the cost can be estimated reliably:
>    - Per-Class Cost (for prototypical concepts $C_p$): Scales linearly ($O(N)$ for $N$ classes). Usage depends on the "Coverage" criterion (iteratively generating concepts until all support samples are covered).
>
>    - Per-Pair Cost (for discriminative concepts $C_d$): Scales quadratically ($O(N^2)$) in an exhaustive setup. However, the unit effort to distinguish a specific pair is constant, governed by a "Validity" criterion (requiring discriminative scores).
>
>    - Tuning Cost: Scales with the number of refinement rounds. While strictly regularized tuning may trigger retries, the cost is bounded by the few-shot size and does not grow indefinitely.
>
> 3. **Quantitative Evidence (API Usage)**: To validate this stability, we experimented on datasets ranging from 2 to 8 classes (SST-2, AGNews, FinancialQueries).
>
>    - First, we provide the Total API Calls per dataset as requested.
>    - Second, we present the Average API Calls Per Class for SST2, AGNews, and FinaQuery. Crucially, these metrics remain consistent across datasets (e.g., $C_d$ cost is stable ~9-10 calls whether $N=2$ or $N=8$). This confirms that our method scales predictably.
>
> *Total API Calls*
> | Dataset | Classes | Total API Calls (10-shot) |
> | :--- | :---: | :---: |
> | SST-2 | 2 | 75 |
> | AGNews | 4 | 173 |
> | FinaQuery | 8 | 1072 |
>
> *Average API Calls Per Class on Datasets*
> |  Per class | 1 shot |  5 shot | 10 shot |
> |---------|--------------| --- | --- |
> | Cp    |         4.83        | 11.12 | 15.38 |
> | Cd    |             4.07   |  9.38  |  9.08 |
> | Tuning    |           0         | 9.77 | 13.01 |
>
>
> 4. **Addressing Bottlenecks ($N \gg 10$)**: We acknowledge that for scenarios with hundreds of classes, the exhaustive $O(N^2)$ generation for $C_d$ becomes the bottleneck. We thus propose the **similarity-based pre-filtering** strategy, whose details are as follows:
>
>    - Mechanism: After generating $C_p$, we use the representations of $C_p$ to evaluate the similarity between classes. Then for each class, its $C_d$ is only generated for its top-$K$ similar classes. This avoids unnecessary comparisons between semantically distant topics (e.g., Sports vs. Finance) and improves efficiency.
>
>    - Related Cost: This strategy reduces the search space from global $O(N^2)$ to $O(NK)$. Because the filtering process does not require LLM calls, the major cost lies in the $C_d$ generation. For $N$ classes, each class should compare with $K$ candidates, which results in the $O(NK)$ complexity.
>
> 5. **Runtime and Memory Footprints**: It is crucial to distinguish between phases:
>    - Runtime: The LLM latency is a one-time offline construction cost. Once the library is built, Online inference is efficient and independent of LLM speed.
>    - Memory: The memory footprint is negligible. Storing embeddings for hundreds of concepts requires <10MB, which is insignificant compared to the pre-trained language model backbone (~500MB). Thus, memory is not a limiting factor for scaling.

---

> ### Author Response · Authors · 2025-11-23
>
> **XLoV-W2, Q2**: It is suggested to estimate the API calls required per dataset. An ablation study on different choices of LLMs is also suggested.
>
> **Reply to XLoV-W2, Q2**:
> We sincerely appreciate the reviewer for the suggestions. We have included the number of API calls in the efficiency table above to better quantify the cost of our method on different datasets. In the following table, our method significantly reduces the number of API calls compared to the baseline LLM method, demonstrating its cost-effectiveness. It is worth noting that our method requires API calls **only during concept construction**, while LLM methods require API calls for **each sample during inference**. Thus, our complexity will not increase with the dataset like LLM baselines, which brings us better scalability.
>
> *API call comparison with LLM*
> | Dataset | Method | API Calls | Accuracy (%) |
> |---|---|---|---|
> | SST2    | LLM (DeepSeek-ICL) |1820| 96.30|
> | SST2    | Ours |75| 80.29|
> | AGNews    | LLM (DeepSeek-ICL) |7600| 79.00|
> | AGNews    | Ours |173| 81.58 |
>
> Additionally, we include an ablation study on different choices of LLMs for concept annotation. Considering the ability of the exsiting LLM ``DeepSeek-V3``, we select ``GPT-4o`` and ``Qwen3-Plus`` for higher-level and lower-level comparisons. The results are listed in the table below. The results indicate that a better LLM generally leads to improved performance, and our method is generally robust to different LLM choices.
>
> *Evaluation on Different LLMs*
> | Dataset | LLM | Accuracy |
> |---------|--------------|---|
> | SST2    | DeepSeek |0.8029|
> | SST2    | GPT-4o |0.8638 |
> | SST2    | Qwen3-Plus | 0.7963 |
> | AGNews    | DeepSeek |0.8158|
> | AGNews    | GPT-4o |0.8507|
> | AGNews    | Qwen3-Plus |0.8176|
>
>
> **XLoV-W3, Q3**: Human evaluation on the quality of the constructed concepts is suggested to better validate the interpretability of the proposed method.
>
> **Reply to XLoV-W3, Q3**:
> We will recruit volunteers to examine 1) whether the annotation from LLM aligns with human and 2) whether the concepts and their related texts match human understanding. This evaluation is currently in process, and the results will be reported within the rebuttal period.
>
> **XLoV-W4**: Add comparison with smaller models that are fine-tuned on few-shot data to better validate the contribution of the proposed method.
>
> **Reply to XLoV-W4**:
> We sincerely appreciate the reviewer for the suggestion. We have included two smaller models ``DistilBert`` and ``RoBERTa`` for comparison. We finetuned both models on few-shot training data, and the results are listed in the table below. The results show that these black-box models perform worse than ours in few-shot settings, demonstrating the effectiveness of our method.
>
> *Comparison with small black-box models*
> | Method | SST2 | MedAbs |
> |---|---|---|
> | RoBERTa    | 50.08 | 51.70 |
> | DistilBert    | 58.59 | 43.49 |
> | StructCBM(Ours)    | 80.29 | 60.11 |
>
> **XLoV-Q4**: The tuning part is essential for the proposed method, it is suggested to provide more detailed ablation study on the semantic tuning and logical tuning stages.
>
> **Reply to XLoV-Q4**:
> We sincerely appreciate the reviewer for the suggestion. We have conducted ablation studies on the semantic tuning and logical tuning stages. The results are presented in the following table. All components do not include embedding model training. It can be seen that both components contribute positively to the final performance, demonstrating the effectiveness of each stage in the tuning process.
>
> *Ablation Study on Concept Tuning*
> | Dataset | Method | Accuracy (%) |
> |---|---|---|
> | AGNews    | No Tuning | 79.30|
> | AGNews    | Semantic Tuning Only | 80.24|
> | AGNews    | Logical Tuning Only | 80.77|
> | AGNews    | Full Tuning |81.41|

---

### Official Review · Reviewer_BfVT · 2025-11-04

**Soundness:** 3
**Presentation:** 3
**Contribution:** 3
**Rating:** 6
**Confidence:** 3

**Summary:**

This paper introduces StructCBM, a novel framework that successfully enables Concept Bottleneck Models (CBMs) to operate in few-shot text classification settings (e.g., 10-shot). The key innovation is a "classifier-free" paradigm that bypasses the need for a data-hungry trainable classifier, which is "impossible" to optimize with limited data. Instead, it relies on direct sample-concept similarity. This is operationalized via a dual-level "prototypical-discriminative" concept architecture and a "generate-predict-refine" loop where an LLM discovers and tunes concepts from a few samples. The method outperforms prior CBMs in the 10-shot regime and achieves performance comparable to black-box LLMs on semantically-rich datasets, all while being interpretable and efficient (LLM-free) at inference.

**Strengths:**

- Novelty: This work successfully and practically applies LLM-augmented CBMs to the few-shot learning domain. It directly addresses a clear gap in the literature, as prior CBMs either required large training sets (like CB-LLM) or costly LLMs at inference (like TBM).
- Strong Empirical Performance vs. Baselines: The model outperforms prior interpretable CBM baselines (TBM, CB-LLM) in the 10-shot setting across all datasets.
- Competitive with Black-Box LLMs: The method achieves performance comparable to, and even exceeding (on AGNews), strong black-box LLM baselines (Deepseek-ICL). It achieves this while being "white-box" interpretable and "lightweight" at inference.
- Sound and Validated Methodology: The dual-level prototypical  and discriminative architecture  is well-motivated for its recall-then-precision task decomposition. This design is empirically validated by both the ablation study (Table 3, showing $\mathcal{C}_d$ is crucial)  and the case study (Fig 4, showing $\mathcal{C}_d$ correcting an initial error).

**Weaknesses:**

- Overfitting: The concept refinement loop, a key part of the method, is shown to be brittle. The ablation study (Table 3) reveals that the full model (with refinement) performs worse than a simpler version on the SST2 dataset, which the authors attribute to overfitting
- Additional Parameters: The "classifier-free" model rests on the quality of a base embedding model (`all-mpnet-base-v2`), which is a critical, unanalyzed hyperparameter. Furthermore, the "Embedding Model Finetuning" step (Sec 3.5)  re-introduces a training phase, which complicates the "classifier-free" narrative and appears responsible for a massive performance jump on FinaQuery (Table 3).

Minor:

- There is some invisible text on Page 15.

**Questions:**

1.  **On Embedding Model Sensitivity:** Your entire method relies on `sim(e(x), e(c))`. How sensitive is the model's performance to the choice of the base embedding model $e(\cdot)$? Why was `all-mpnet-base-v2` chosen, and would a different model change the results?
2.  **On Refinement Overfitting:** The refinement (Alg 1) appears to overfit on SST2. This seems due to the greedy, 1-sample error correction objective. Did you experiment with a more regularized tuning objective (e.g., one that forces a concept to remain similar to *all* its correct samples while becoming dissimilar to the one error sample)?
3.  **Finetuning** The ablation study (Table 3)  shows that the "Train" step (Sec 3.5) provides a significant boost, especially on FinaQuery. How much does this finetuning of the embedding model contribute to the final performance, and does this reliance on a training step weaken the "classifier-free" premise?
4.  **LLM Annotation Reliability:** The "Concept Annotation and Selection" step (Fig 2)  uses an LLM to score logical consistency. How reliable is this process? What is the impact of potential LLM annotation errors on the final concept library and performance?

---

> ### Author Response · Authors · 2025-11-23
>
> We thank the reviewer for the effort in reviewing our paper. We appreciate the recognition of our **novelty, strong empirical performance, and sound and validated method**. Our replies to the concerns are presented below. We will prepare a revised version of the paper accordingly.
>
> ---
>
> **BfVT-W1, Q2**: The refinement on SST2 appears to introduce the overfitting problem, which makes the refinement process brittle. It is suggested to solve such an issue by adding regularization objectives.
>
> **Reply to BfVT-W1, Q2**:
> We sincerely appreciate the reviewer for accurately analyzing our algorithm. As per your suggestion, we have incorporated regularization objectives into our current framework. Specifically, during the refinement phase for Discriminative Concepts ($C_d$), we introduced a stricter constraint: in addition to distancing the concept from the negative samples, we now enforce that the concept's average similarity to all the positive samples must not decrease. This guarantees the minimum generalization ability of the refined concepts to the known samples.
>
> We evaluate the effect of regularization on SST2 dataset across multiple iterations and present the results in the table below. As hypothesized and demonstrated in the table, by enforcing the average-similarity constraint (w/ Reg.), the model achieves stability and reaches a peak accuracy of 81.27% on SST2 dataset, surpassing the results with no refinement.
>
> *Effectiveness of Regularization*
> | Dataset | Refinement Method | Accuracy (%) |
> |---|---|---|
> | SST2  | No Refine | 80.29         |
> | SST2  | Refine w/o Reg. |  76.77    |
> | SST2  | Refine w/ Reg. | 81.27      |
>
> We further analyze that the overfitting tendency on SST2 is caused by a structural nuance in binary classification. Our Discriminative Concepts ($C_d$) were originally designed to strictly reject a confusing class without necessarily supporting the target class. However, in the strictly dichotomous setting of SST-2, a concept that rejects 'negative' is semantically coupled with supporting 'positive', meaning these concepts must possess clear polarities. Consequently, our original unconstrained refinement could inadvertently sever this crucial alignment with the positive class. This finally makes regularization necessary.
>
> **BfVT-W2, Q1**: The embedding model plays a critical role in the proposed method, it is suggested to explain the selection of existing embedding model and provide sensitivity analysis on how different embedding models affect the final performance.
>
> **Reply to BfVT-W2, Q1**:
> We acknowledge that the choice of the foundation embedding model is pivotal. We selected ``all-mpnet-base-v2`` primarily because it is widely recognized as a robust sentence embedding model for semantic similarity tasks. Furthermore, this specific architecture is also utilized by recent relevant works, such as CB-LLM [1], which allows for a fair and consistent comparison with established baselines.
>
> Following your suggestion, we conducted a sensitivity analysis to examine how different embedding models affect final performance. We select a better model ``Qwen3-Embedding-0.6B`` and a worse model ``all-MiniLM-base-v6`` for thorough comparisons. The results in the table below indicate that a better embedding model leads to improved performance, and our method is generally robust to different embedding models.
>
> | Dataset | Embedding Model | Accuracy (%) |
> |---|---|---|
> |AGNews| Qwen3-embedding-0.6B          | 86.50|
> |AGNews| all-mpnet-base-v2          | 81.58|
> |AGNews| all-MiniLM-L6-v2          | 80.17|
> |MedAbs| Qwen3-embedding-0.6B          | 61.43|
> |MedAbs| all-mpnet-base-v2          | 60.11|
> |MedAbs| all-MiniLM-L6-v2          | 56.68|

---

> ### Author Response · Authors · 2025-11-23
>
> **BfVT-W2, Q3**: Finetuning the embedding model brings significant improvements; it is suggested to clarify its relationship with the claim of "Classifier-free" setting and quantify its contribution.
>
> **Reply to BfVT-W2, Q3**: We would like to clarify that finetuning the embedding model does not diminish our "classifier-free" advantages. The primary motivation behind our approach is to address the inherent instability of training classifiers in few-shot scenarios, rather than reducing training costs, which is often the focus of "training-free" methods.
>
> Specifically, by "classifier-free," we refer to the elimination of the traditional "Concept-to-Label mapping layer," denoted as $g(c) \rightarrow y$, which is a staple in standard Concept Bottleneck Models (CBMs). In standard CBMs, $g$ functions as a trainable classifier (e.g., a linear layer) where weights represent the correlation between concepts and labels. Given that the concept space is often vast (typically $10^1$ to $10^2$ concepts per class), training $g$ requires substantial data. In a few-shot setting (e.g., 10-shot), there are insufficient samples to learn these weights effectively, which is the core reason why traditional CBMs struggle in such scenarios.
>
> To resolve this, we remove $g$ entirely. In our framework, the relationship between concepts and labels is governed by hard logical rules (e.g., if a concept is selected, the label is deterministically decided). Consequently, our optimization task simplifies from training a classifier to matching samples with concepts. We utilize a sentence embedding model to perform this matching.
>
> While the model performs well even in its raw state, we seek to maximize performance for specific tasks through vertical domain adaptation. We fine-tune the embedding model using sample-concept pairs strictly to align the general semantic space of the pre-trained model with the specific nuances of the vertical domain, ensuring more accurate similarity calculations, rather than to train a classifier.
>
> Following your suggestion, we conducted a sensitivity analysis to observe how different embedding models affect final performance. We tested various models before and after the fine-tuning process. As shown in the additional experiments, performance generally positively correlates with the capabilities of models. Furthermore, fine-tuning yields consistent improvements across all tested architectures, demonstrating that our method benefits robustly from better semantic representations.
>
> *Contribution of Finetuning*
> | Dataset | Embedding Model | Pre-Finetuning Accuracy (%) | Post-Finetuning Accuracy (%) |
> |---|---|---|---|
> |AGNews| Qwen3-embedding-0.6B | 80.49 | 86.50 |
> |AGNews| all-mpnet-base-v2 | 81.41 | 81.58 |
> |AGNews| all-MiniLM-L6-v2 | 75.04 | 80.17 |
> |MedAbs| Qwen3-embedding-0.6B | 58.75 | 61.43 |
> |MedAbs| all-mpnet-base-v2 | 59.00 | 60.11 |
> |MedAbs| all-MiniLM-L6-v2 | 55.47 | 56.68 |
>
> **BfVT-Q4**: How reliable is the "Concept Annotation and Selection" process? What is the impact of potential LLM annotation errors on the final performance?
>
> **Reply to BfVT-Q4**: To rigorously quantify the reliability and robustness of the LLM's judgments, we designed two supplementary experiments to assess both the **internal consistency of the LLM** and its **external validity in alignment with human reasoning**:
>
> 1. **Internal Consistency Test**: We validate the reliability of LLM by examing whether it can remain consistant when annotating the same concepts for multiple independent trials. Similar ideas have been used in recent works to examine the reliability of LLMs, such as Self-Refine [2] and Reflexion [3]. We randomly sampled 20 concepts and their corresponding texts from two datasets. We then queried the LLM to score these pairs 10 times independently. On our hardest dataset MedAbs, the experiment revealed a high consistency on **98.6%** of all trails. This confirms that the LLM operates with a stable internal logic and is consistent in its filtering criteria.
>
> 2. **External Validity Test (Human-AI Alignment)**: To ensure that the LLM's internal logic aligns with human reasoning, we initiated a human evaluation study. We will recruit volunteers to examine 1) whether the annotation from LLM aligns with human and 2) whether the concepts and their related texts match human understanding. This evaluation is currently in process, and the results will be reported within the rebuttal period.
>
> **BfVT-Minor**: There is some invisible text on Page 15.
>
> **Reply to BfVT-Minor**: We have reviewed the document and will correct the invisible text on Page 15 to ensure clarity and readability. In the submitted version, the invisible text is actually the text from Fig. 6 and is harmless.
> Thank you for bringing this to our attention.
>
> - [1] ICLR 2025 Concept bottleneck large language models
> - [2] Self-Refine: Iterative Refinement with Self-Feedback
> - [3] Reflexion: Language Agents with Verbal Reinforcement Learning

---

### Author Response · Authors · 2025-12-03
**General Response: Human Evaluation Results**

Multiple reviewers raised concerns regarding the reliability of utilizing LLMs for concept generation and annotation. Thus, we conducted human evaluations to address the concerns from two dimensions: (1) Contribution Faithfulness (Do the concepts explain the prediction?) and (2) Annotation Reliability (Is the LLM judgment trustworthy?). Our model achieved a **Contribution Faithfulness score of 87.8% (4.39/5.0)** and an **Annotation Reliability agreement of 96.43%** with human consensus. These results empirically confirm that our LLM-driven process aligns closely with expert reasoning and produces trustworthy explanations.

 **Experimental Setup** To ensure diversity, we randomly picked 2 training and 2 testing samples from each class across the SST-2, AGNews, and FinaQuery datasets, resulting in 56 samples in total. Given the domain-specific complexity of the FinaQuery dataset, we recruited three independent evaluators with academic backgrounds in Finance to guarantee high-quality judgment on concept validity.

- **Contribution Faithfulness** Task: We presented evaluators with samples drawn explicitly from the test sets of our datasets. For each instance, evaluators reviewed the input text alongside the model’s predicted label and the Top-3 concepts utilized by StructCBM. They were then asked to rate, on a 5-point Likert scale, whether these concepts provided sufficient evidence to justify the classification. StructCBM achieved a high average rating of **4.39/5.0**, strongly indicating that the identified concepts offer semantically meaningful and sufficient explanations that align with human expert reasoning.

- **Annotation Reliability** Task: To validate the reliability of the "Concept Annotation" phase described in Section 3.2, we conducted a blind agreement test on the training samples where evaluators manually determined the relatedness of specific concepts in the text to establish a "Human Ground Truth." Comparing these manual annotations against the LLM's automated scoring, we found that the LLM matched the human expert consensus in 27 out of 28 cases, achieving an agreement rate of **96.43%**. This high level of concordance empirically validates our reliance on LLMs for feature annotation, demonstrating that they operate with precision comparable to human domain experts in this setting.


Collectively, these results confirm that StructCBM produces highly interpretable outputs underpinned by a robust and reliable LLM-driven annotation process.

---

### Author Response · Authors · 2025-12-03
**Response Summary (3/3)**

### 4. Demonstrated Robustness (LLMs, Embeddings, & Shots)
We conducted extensive sensitivity analyses to confirm that our framework is robust and not dependent on a specific configuration.

**A. LLM Dependency & Consistency**

- **Internal Consistency Test**: On our most challenging dataset (MedAbs), the LLM maintained **98.6% consistency across 10 independent annotation trials**, confirming stable internal logic.

- **Ablation on Different LLMs**: Results show robustness across LLM choices

*Table 3: Evaluation on Different LLMs*

| Dataset | DeepSeek-V3 | GPT-4o | Qwen3-Plus |
| ------- | ----------- | ------ | ---------- |
| SST2    | 80.29%      | 86.38% | 79.63%     |
| AGNews  | 81.58%      | 85.07% | 81.76%     |

**B. Embedding Model Analysis**

We validated performance across different embedding architectures. Fine-tuning consistently improves performance across all architectures, demonstrating robust benefits from better semantic representations.

*Table 4: Evaluation on Different embedding models*

| Dataset | all-MiniLM-L6-v2 | all-mpnet-base-v2 | Qwen3-embedding-0.6B |
| ------- | ---------------- | ----------------- | -------------------- |
| AGNews  | 80.17%           | 81.58%            | 86.50%               |
| MedAbs  | 56.68%           | 60.11%            | 61.43%               |

**C. Robustness Across Few-Shot**

We verified consistent gains across 1-shot, 5-shot, and 10-shot settings, indicating strong few-shot robustness.

*Table 5: Experiments on different few-shot settings*
| Dataset        | 1-shot | 5-shot | 10-shot |
| -------------- | ------ | ------ | ------- |
| AGNews         | 65.71% | 80.76% | 81.58%  |
| FinancialQuery | 58.73% | 73.60% | 77.42%  |



### 5. Strengthened Baselines & Conceptual Clarity

**A. New Baseline Comparisons**

We added comparisons with white-box models and fine-tuned small models, confirming StructCBM’s superiority in data-scarce regimes.

- **Additional White-Box Baselines** (C3M, SparseCBM):

*Table 6: Comparison with additional white-box baselines*

| Method          | MedAbs | AGNews |
| --------------- | ------ | ------ |
| C3M             | 31.09% | 55.68% |
| SparseCBM       | 37.78% | 62.84% |
| StructCBM(Ours) | 60.11% | 81.58% |

- **Small Black-Box Models** (fine-tuned on few-shot data):

*Table 7: Comparison with small black-box models*
| Method          | SST2   | MedAbs |
| --------------- | ------ | ------ |
| RoBERTa         | 50.08% | 51.70% |
| DistilBert      | 58.59% | 43.49% |
| StructCBM(Ours) | 80.29% | 60.11% |

These results confirm that data scarcity is the primary challenge, and our method effectively addresses it.

**B. Conceptual Clarifications**

- **"Classifier-Free" Definition**: We clarified that our definition of "classifier-free" means our framework structurally eliminates the data-hungry parametric mapping (weights) between concepts and labels, distinguishing it from traditional CBMs that require extensive training.

- **Component Ablation**: We evaluated the contribution of each tuning component:

*Table 8: Ablation Study on Concept Tuning*

| Dataset | No Tuning | Semantic Only | Logical Only | Full Tuning |
| ------- | --------- | ------------- | ------------ | ----------- |
| AGNews  | 79.30%    | 80.24%        | 80.77%       | 81.41%      |


- **Prototypical vs. Discriminative Concepts**: We provided concrete examples from the FinancialQuery dataset illustrating how prototypical concepts capture universal commonalities while discriminative concepts act as exclusionary filters.

- **Fine-tuning Dataset Construction**: Adding $C_d$ to fine-tuning data provides additional improvements, though subtle compared to $C_p$:

*Table 9: Contribution of Finetuning*

| Dataset | None   | Only $C_p$ | $C_p$ + $C_d$ |
| ------- | ------ | ---------- | ------------- |
| AGNews  | 81.41% | 81.58%     | 82.05%        |
| MedAbs  | 59.00% | 60.11%     | 60.70%        |



## Conclusion

We believe our responses comprehensively address all reviewer concerns and provide substantial additional evidence supporting our claims. We will incorporate all suggested changes and additional experiments into the revised manuscript. We remain committed to addressing any further questions during the discussion period.

---

### Author Response · Authors · 2025-12-03
**Response Summary (2/3)**

## Detailed Summary

### 1. Resolved Overfitting via Strict Regularization

We implemented a stricter regularization constraint during the refinement phase for Discriminative Concepts ($C_d$). Specifically, we enforce that the concept's average similarity to positive samples must not decrease while being adjusted. This successfully stabilized the tuning process:

*Table 1: Effectiveness of Regularization*

| Dataset | No Refine | Refine w/o Reg. | Refine w/ Reg. |
| ------- | --------- | --------------- | -------------- |
| SST2    | 80.29%    | 76.77%          | **81.27%**     |

Mechanism Analysis: We attribute the instability to the unique nature of binary classification. In the strictly binary SST-2 dataset, a concept designed to reject 'negative' is semantically coupled with supporting 'positive'. This contrasts with multi-class settings, where rejecting one label (e.g., "Student") does not automatically imply support for another specific label (e.g., "Teacher"). Our updated regularization effectively addresses this by enforcing that the concept must remain aligned with the positive samples.


### 2. Validated Scalability & Efficiency

We provided a comprehensive analysis including unit costs, total API usage, stopping criteria, and future strategies to demonstrate scalability:



**A. Stable Unit Costs** We demonstrated that our unit costs remain stable across datasets, proving predictability:

**Average API Calls Per Unit (10-shot)**:
- $C_p$ generation: ~15.38 calls
- $C_d$ generation: ~9.08 calls
- Tuning: ~13.01 calls


Note on Cost Composition: It is important to clarify that these estimates encompass the entire **Generation-Validation-Retry loop**.

- Generation ($C_p/C_d$): The $\sim$15.38 calls for $C_p$ reflect our rigorous process to meet stopping criteria (e.g., ensuring 100% sample coverage). We prioritize concept quality over minimal API usage to ensure a robust foundation.
- Tuning Phase: The tuning cost is inherently dynamic, correlating with the number of errors corrected. Crucially, this represents a configurable trade-off. While we allowed extensive refinement rounds in this study to maximize few-shot accuracy (Performance Mode), in practical deployment, users can impose stricter limits on tuning iterations (e.g., capping at 2 rounds) to significantly reduce costs with minimal impact on final performance.

For a detailed breakdown of stopping criteria and convergence analysis, please refer to our Reply to swGd-W4.



**B. Total API Call Comparison & Efficiency vs. LLMs**

We quantified total API costs to demonstrate scalability. As shown below, StructCBM significantly reduces API usage compared to LLM baselines. Crucially, our method incurs API costs only once during offline construction, so our cost **does not increase with test data volume**, ensuring superior scalability.

*Table 2: API call comparison with LLM*
| Dataset | Method | API Calls | Accuracy (%) |
|---|---|---|---|
| SST2    | LLM (DeepSeek-ICL) |1820| 96.30|
| SST2    | Ours |75| 80.29|
| AGNews    | LLM (DeepSeek-ICL) |7600| 79.00|
| AGNews    | Ours |173| 81.58 |


**C. Strategy for Large-Scale Scenarios** For scenarios with hundreds of classes, we proposed a **similarity-based pre-filtering strategy**:
- Reduces complexity from $O(N^2)$ to $O(NK)$ by generating $C_d$ only for top-K similar classes
- Avoids unnecessary comparisons between semantically distant classes
- Memory footprint remains negligible (<10MB for concept embeddings) even at scale.


### 3. Confirmed Interpretability & Reliability (Human Eval)

To rigorously validate concept quality, we recruited three independent evaluators with academic backgrounds in Finance to assess a total of 56 stratified samples (28 for validity, 28 for reliability).

- **Validity (Contribution Faithfulness)**: Evaluators rated whether the Top-3 concepts provided sufficient evidence for the prediction on a 5-point scale. StructCBM achieved a high average rating of **4.39 / 5.0**, confirming the concepts serve as meaningful explanations.

- **Reliability (Annotation Consistency)**: In a blind consistency test, the LLM matched human expert consensus in **96.43%** of cases (27/28 matches), empirically proving its trustworthiness.

- **Quality Control Mechanisms**: Our framework actively mitigates unreliable concepts through (1) LLM annotation filtration at the beginning and (2) Concept tuning to align representations with actual data.

---

### Author Response · Authors · 2025-12-03
**Response Summary (1/3)**

Dear Area Chair,

Thank you for your efforts in reviewing our paper and rebuttal content. Here, we present a summary of the concerns from reviewers and our responses.

**Summary of Contributions**: This paper targets a primary difficulty in the few-shot text classification problem: while black-box LLMs offer high performance, they lack efficiency and interpretability; conversely, traditional Concept Bottleneck Models (CBMs) offer transparency but fail to train effectively with limited samples. To resolve this, we introduce StructCBM, which employs a novel dual-concept architecture with adaptive mechanisms to enable a classifier-free inference structure. By bypassing the need for classifier training, our approach successfully reconciles interpretability with LLM-approaching performance in data-scarce regimes.

**Brief Response Summary**

We sincerely thank all reviewers for their thoughtful and constructive feedback. We are encouraged by the recognition of our work's **novelty, clear presentation, strong empirical performance, and high interpretability**. To address the questions raised, we have conducted extensive new experiments and analyses. A summary of the key concerns and our corresponding responses follows below:

**(1) Resolved Overfitting via Strict Regularization**:

`Corresponding to BfVT-W1, Q2, GAko-Q2, mVk8-W1, Q2`

`Revised Section 3.4 and 4.3`

Addressing concerns on SST-2 brittleness, we implemented a stricter regularization that anchors discriminative concepts to positive samples. This resolved the overfitting caused by the binary classification structure, improving SST-2 refinement accuracy from 76.77% to 81.27%.

**Global Performance Update**: Encouraged by the improvement on SST-2, we applied the optimized regularization strategy to all datasets for consistency. This yielded performance gains across most benchmarks (e.g., AGNews +3.87%, MedAbs +0.59%), which are now reflected in the revised paper. Please note that this update strengthens our results without affecting the validity of the conclusions drawn in other parts of this rebuttal.

**(2) Validated Scalability & Efficiency**:

`Corresponding to BfVT-W1, Q2, XLoV-W1, W2, Q1, Q2, swGd-W1, W4, GAko-W1, Q1`

`Revised Section 4.5 and Appendix.A.8`

-  Stable Unit Cost & Efficiency: We provided a detailed cost breakdown proving that offline construction follows a stable unit-cost model (e.g., ~15 calls per $C_p$, ~9 calls per $C_d$ pair for 10-shot, decreasing for fewer shots). Crucially, online inference is API-free with zero overhead.

-  Scalability Strategy: Addressing scenarios with hundreds of classes ($N$), we proposed a pruning strategy that clusters semantically similar labels into groups ($K$). This reduces the pairwise generation complexity from $O(N^2)$ to $O(NK)$, ensuring scalability.



**(3) Confirmed Interpretability & Reliability (Human Eval)**:

`Corresponding to BfVT-Q4, swGd-W3, mVk8-W1, Q2`

`Revised Section 4.6 and Appendix.A.7`

We conducted expert-level human evaluations. Results show high Contribution Faithfulness (4.39/5, where 5=Perfect Evidence) and exceptional Annotation Agreement (96.43%), empirically validating that our LLM-driven concepts are trustworthy and align with human expert reasoning.


**(4) Demonstrated Robustness (LLMs, Embeddings, & Shots)**:

`Corresponding to BfVT-W2, Q1, XLoV-W3, Q3, swGd-W2, W3, GAko-W2`

`Revised Section 4.4`

- LLM/embedding Dependency: We confirmed robustness across various LLMs (GPT-4o, Deepseek, Qwen) and embedding models (mpnet, MiniLM, Qwen3-embeddding-0.6B).
- Few-Shot Settings: We verified consistent performance gains across 1-shot, 5-shot, and 10-shot settings.


**(5) Strengthened Baselines & Conceptual Clarity**:


`Corresponding to BfVT-W2, Q3, XLoV-W4, mVk8-W2, W4, Q3`

`Revised Section 3.1, 4.2`

- New Baselines: We added comparisons with white-box models (C3M, SparseCBM) and fine-tuned small models (RoBERTa, DistilBERT), confirming StructCBM’s superiority in data-scarce regimes.
- "Classifier-Free" Definition: We clarified that our framework removes the learnable weights between concepts and labels, avoiding the need for large-scale data typically required to train this mapping in traditional CBMs.

---

### Meta-Review · Area_Chair_cweN · 2025-12-30

**Summary:**

This paper presents an LLM-augmented Concept Bottleneck Model (CBM), named StructCBM, for interpretable few-shot text classification, which leverages an adaptive concept library. StructCBM predicts labels through sample–concept similarity, removing the need for parametric training. The authors also introduced a dual-level concept structure, i.e., prototypical concepts (Cp) capture representative class features for recall, while discriminative concepts (Cd) distinguish confusable classes for precision. Extensive results on multiple benchmark datasets are reported and discussed.

Reviewers agreed that this paper studies an important problem and explores an under-explored intersection of interpretability and few-shot learning. The proposed StructCBM method is novel and technically sound. The proposed two-stage approach is well motivated and clearly explained. Experimental results are comprehensive and convincing. In addition, the paper is well written and easy to follow.

Meanwhile, reviewers raised some concerns about overfitting, reliability, efficiency, baselines, and experimental settings.

**Reviewer Concerns:**

The authors have provided very detailed responses and additional results, which have addressed most of the concerns from reviewers. The  new results and discussions have been incorporated into the revised paper.

**Reviewer Scores:**

Initially this paper received borderline ratings: 6, 6, 6, 4, and 4. Considering that most of the concerns have been well addressed in the rebuttal, I think at least one of the reviewers would have changed their score from 4 to 6. Thus, the overall assessment of this paper is positive.

---

### Decision · Program_Chairs · 2026-01-26

Accept (Poster)